



# Diel quenching of Southern Ocean phytoplankton fluorescence is related to iron limitation

Christina Schallenberg[1], Robert F. Strzepek[1], Nina Schuback[2], Lesley A. Clementson[3], Philip W. Boyd[1,4], Thomas W. Trull[1,3]

[1]Antarctic Climate and Ecosystems Cooperative Research Centre, University of Tasmania, Hobart, Tasmania, Australia
[2]Swiss Polar Institute, École Polytechnique Fédérale de Lausanne, Lausanne, Switzerland
[3]Commonwealth Scientific and Industrial Research Organisation Oceans and Atmospheres, Hobart, Tasmania, Australia
[4]Institute for Marine and Antarctic Studies, University of Tasmania, Hobart, Tasmania, Australia

*Correspondence to*: Christina Schallenberg (christina.schallenberg@utas.edu.au)
ORCID: 0000-0002-3073-7500

**Abstract.**

Evaluation of photosynthetic competency in time and space is critical for better estimates and models of oceanic primary productivity. This is especially true for areas where the lack of iron limits phytoplankton productivity, such as the Southern

Ocean. Assessment of photosynthetic competency on large scales remains challenging, but phytoplankton chlorophyll-a fluorescence (ChlF) is a signal that holds promise in this respect as it is affected by, and consequently provides information about, the photosynthetic efficiency of the organism. A second process affecting the ChlF signal is heat dissipation of absorbed light energy, referred to as non-photochemical quenching (NPQ). NPQ is triggered when excess energy is absorbed; i.e., when more light is absorbed than can be used directly for photosynthetic carbon fixation. The effect of NPQ

on the ChlF signal complicates its interpretation in terms of photosynthetic efficiency, and therefore most approaches relating ChlF parameters to photosynthetic efficiency seek to minimize the influence of NPQ by working under conditions of sub-saturating irradiance. Here, we propose that NPQ itself holds potential as an easily acquired optical signal indicative of phytoplankton physiological state with respect to iron (Fe) limitation.

We present data from a research voyage to the Subantarctic Zone south of Australia. Incubation experiments confirmed that resident phytoplankton were Fe-limited, as the maximum quantum yield of primary photochemistry, Fv/Fm, measured with a Fast Repetition Rate fluorometer (FRRf), increased significantly with Fe addition. The NPQ "capacity" of the phytoplankton also showed sensitivity to Fe addition, decreasing with increased Fe availability, confirming previous work. The fortuitous presence of a remnant warm-core eddy in the vicinity of the study area allowed comparison of fluorescence behaviour

between two distinct water masses, with the colder water showing significantly lower Fv/Fm than the warmer eddy waters, suggesting a difference in Fe limitation status between the two water masses. Again, NPQ capacity measured with the FRRf mirrored the behaviour observed in Fv/Fm, decreasing as Fv/Fm increased in the warmer water mass. We also analysed the diel quenching of underway fluorescence measured with a standard fluorometer, such as is frequently used to monitor ambient chlorophyll-a concentrations, and found a significant difference in behaviour between the two water masses. This

difference was quantified by defining an NPQ parameter akin to the Stern-Volmer parameterization of NPQ, exploiting the fluorescence quenching induced by diel fluctuations in incident irradiance. We propose that monitoring of this novel NPQ parameter may enable assessment of phytoplankton physiological status (related to Fe availability) based on measurements made with standard fluorometers, as ubiquitously used on moorings, ships, floats and gliders.

**1. Introduction**

A key limitation to confident estimates of global ocean productivity is the lack of readily obtained information regarding the physiological status of phytoplankton. Assessment of photosynthetic competency in time and space is a crucial requirement





for improved estimates and models of oceanic primary productivity. This is especially true for areas where iron (Fe)

limitation is prevalent, such as the Southern Ocean (Boyd et al., 2007; Moore et al., 2013). Fe-induced variations in primary

productivity in this region have been shown to arise from physiological drivers of photosynthesis that are currently poorly

represented in models of productivity (Hiscock et al., 2008). The Southern Ocean is of particular interest because of its large

influence on the global carbon cycle, which is directly linked to photosynthetic performance of the resident phytoplankton

(Martínez-García et al., 2014; Sigman and Boyle, 2000). While photosynthetic performance can readily be measured on

small scales, i.e. during ship-based surveys conducting incubations for estimates of primary productivity, assessment of

photosynthetic competency on larger scales remains challenging. Fluorescence emitted by phytoplankton upon absorption of

light is a signal that holds great promise in this respect, as it stems directly from the photosynthetic apparatus of

phytoplankton and can be measured by instruments mounted on drifters, floats, gliders and even satellites (Letelier et al.,

1997; Behrenfeld et al., 2009; Huot et al., 2013; Morrison, 2003; Schallenberg et al., 2008). Indeed, the signal has been

shown to hold the potential for providing information on the physiological state of phytoplankton (Letelier et al., 1997;

Behrenfeld et al., 2009; Morrison and Goodwin, 2010; Schallenberg et al., 2008).

Chlorophyll-a fluorescence (ChlF) is one of three pathways that light energy can take once absorbed by the light-harvesting

antenna of photosystem II (PSII) of a photosynthetic organism. The other two possible pathways are photochemistry (i.e.,

primary charge separation in reaction center II (RCII)), and heat dissipation. Heat dissipation of absorbed light energy can be

upregulated in situations where light energy is absorbed in excess of photosynthetic capacity, which has the potential to

damage the photosynthetic apparatus. As the three possible pathways of absorbed energy are complementary, an

upregulation of heat dissipation will quench the ChlF signal, which is known as non-photochemical quenching (NPQ)

(Horton, Ruban, and Walters, 1996; Krause and Weis, 1984; Müller et al., 2001).

At midday, NPQ can depress the ChlF signal by up to 90%. This affects day-time ChlF data from ships, satellites, bio-optical

floats and gliders, the latest and rapidly growing additions to the oceanographic toolbox (Biermann et al., 2015; Grenier et

al., 2015; Xing et al., 2012; 2018). Sensitivity of NPQ to the light acclimation state of phytoplankton has been demonstrated

(Milligan et al., 2012; O'Malley et al., 2014), and other drivers of NPQ, including nutrient status and species composition,

have also been suggested (e.g., Kropuenske et al., 2009; Schallenberg et al., 2008; Schuback et al., 2015). An empirical

relationship between sea surface temperature and NPQ in the Southern Ocean has been described (Browning et al., 2014),

but the underlying controls are not fully understood.

Active fluorometers such as a Fast Repetition Rate fluorometer (FRRf) exploit the complementary nature of the three

possible pathways of absorbed light energy. Triggering and detecting changes in ChlF in the dark-regulated state (i.e., when

NPQ is relaxed and does not quench ChlF), allows derivation of the commonly used parameter $F_v/F_m$. $F_v/F_m$, often referred

to as the maximum quantum yield of PSII, is a measure of the maximum fraction of absorbed light energy which can be used

for primary charge separation in RCII. Strong links have been established between phytoplankton Fe status and $F_v/F_m$. As

Fe becomes more and more limiting, $F_v/F_m$ decreases (Geider et al., 1993; Greene et al., 1992). Benchtop FRRf instruments

furthermore can quantify the induction of NPQ in response to increases in absorbed light energy, and recent studies have

found a strong link between the capacity to induce NPQ and the Fe limitation status of phytoplankton (Alderkamp et al.,

2012; Schuback et al., 2015; Schuback and Tortell, 2019).

The term "non-photochemical quenching" is a blanket term for a number of processes and parameterizations, with whole

books dedicated to its many manifestations (e.g., Demmig-Adams et al., 2014). At its core, it is a photoprotective mechanism

that safely removes excess excitation energy from the light harvesting system of photosynthesizers (Müller et al., 2001). It



can be assessed in many ways, the most clearly defined ones measured with active fluorometers such as FRRf, which allow measurements in both the dark- and light-regulated state. Common parameterizations of NPQ include the Stern-Volmer (SV) and Normalized Stern-Volmer (NSV) parameterizations, but a myriad of other definitions are in use (e.g., Rohacek, 2002). In a less strictly defined manner, NPQ can also be detected in measurements that are made with what we will call "standard

fluorometers" (SF) in this manuscript; i.e., fluorometers conventionally employed on floats, gliders, moorings, and ships to estimate ambient chlorophyll-a concentrations as a proxy for phytoplankton biomass (e.g., Roesler et al., 2017). Here, NPQ can take the form of depressed fluorescence in the middle of the day relative to the night, or it can manifest as a depression of fluorescence towards the ocean surface in daytime fluorescence profiles (Biermann et al., 2015; Grenier et al., 2015; Xing et al., 2012; 2018). Standard fluorometers are deployed ubiquitously in the oceans and can be operated remotely; the

prospect of harnessing physiological information contained in the NPQ signal they detect is thus tantalizing. Given the link between NPQ – as estimated with active fluorometers – and the Fe limitation status of phytoplankton (Alderkamp et al., 2012; Schuback et al., 2015; Schuback and Tortell, 2019), we investigated whether the NPQ signal detected by a standard fluorometer can also be interpreted with respect to the Fe limitation status of the resident phytoplankton community.

The two main objectives of this study were as follows: i) Link NPQ capacity as measured with a FRRf to NPQ measured with a standard fluorometer; ii) Link the NPQ signals from both to the Fe limitation status of the resident phytoplankton community. The first objective accounts for the fact that measurements with an FRRf are made under very controlled conditions, while the fluorescence detected with a standard fluorometer is measured under ambient conditions that can be highly variable due to differences in incident sunlight, in the rate of change in illumination (e.g., due to passing clouds), and

in the mixing depth, which also affects light history and acclimation. We further note that there is a ship-specific period of "dark-acclimation" prior to measurement if the standard fluorometer is installed on a ship's underway seawater line, yielding an acclimation status of the phytoplankton that is neither fully dark-regulated, nor fully light-regulated since some NPQ components can be reversed on the order of minutes and even seconds (Müller et al., 2001).

In order to investigate our research objectives, we undertook the following measurements and experiments on a voyage to the Subantarctic Zone (SAZ) south of Australia: 1) deckboard incubation experiments under controlled Fe conditions to confirm the link between Fe limitation and NPQ capacity, as measured with an FRRf, in the resident phytoplankton. 2) FRRf measurements on samples from the underway seawater line, yielding estimates of Fv/Fm and NPQ capacity that could be related to the Fe limitation status of the phytoplankton. This step revealed that two distinct water masses had been repeatedly

sampled by the ship, with significantly different Fv/Fm, indicating differences in their Fe limitation status. 3) NPQ estimated from the standard fluorometer on the underway seawater line was linked to the different water masses and thus to Fe limitation status, as well as to Fv/Fm and NPQ capacity measured with an FRRf.

## 2. Methods


### 2.1 Oceanographic setting and sampling

A research voyage to the Southern Ocean Time Series (SOTS) mooring site near 140°E and 47°S took place from March 3 to 20, 2018 aboard the *RV Investigator*. The SOTS site lies in the SAZ southwest of Tasmania, an area of considerable mesoscale and sub-mesoscale variability (Shadwick et al., 2015; Weeding and Trull, 2014). During occupation of the area, a

remnant eddy with sea surface temperatures (SST) >14°C was just east of the SOTS site and was visited multiple times by the ship (Fig. 1).





The ship's underway seawater supply, which had an intake in the ship's drop keel (water depth ~7 m), was equipped with a WETStar fluorometer (Wetlabs, Inc., Philomath, Oregon, USA) that measured ChlF at 695 nm (excitation at 460 nm)

continuously throughout the voyage. Travel time through the ship's plumbing from intake to the fluorometer was between 1 and 2 minutes. The underway seawater line also supplied water to a WETLabs absorption and attenuation meter (ac-9) in flow-through mode (see below). At regular intervals, the underway seawater supply was sampled for pigment analysis by high performance liquid chromatography (HPLC), particulate absorption spectra following the filter pad approach (Mitchell, 1990) and chlorophyll-a concentration analysed fluorometrically (Chl-a; more details on the respective methods below).

Oceanic profiles (conductivity, temperature, depth, hereafter CTD) were collected at 6 stations using Sea-Bird SBE 9*plus* instrumentation, with associated sampling conducted from a 36-bottle rosette equipped with 12 L Niskin bottles. Onboard measurements of oxygen and salinity on discrete samples from the Niskin bottles were used to calibrate CTD sensors. Samples for macronutrients ($NO_x$ (sum of nitrate and nitrite), Silicate, Phosphate) were also drawn from Niskin bottles and analysed onboard with a Seal AA3 segmented flow instrument following the methods of Rees et al. (2019).


## 2.2 Incubation experiments

At 3 stations, all located in close vicinity to 47°S, 142°E, either a trace metal clean rosette system (TMR) or a trace metal clean pumped "fish" (i.e., a submerged body towed from abeam and outside the ship wake with an intake at 3-5 m depth) were deployed in order to retrieve uncontaminated samples. For each of the 3 incubation experiments, one initial HPLC

sample was drawn, and 4 acid-washed polycarbonate bottles (250 mL each) were filled with unfiltered seawater from the mixed layer and incubated in a deck incubator that was continuously supplied with surface seawater for temperature control. Light levels were controlled with mesh bags placed around the bottles (see Table 1). Each experiment consisted of the following 4 treatments (final concentrations indicated):

- 20 nM desferrioxamine B (DFB), a strong iron chelator

- Control (no additions)

- 0.2 nM Fe (added as $FeCl_3$ dissolved in acidified Milli-Q water)

- 2 nM Fe (added as $FeCl_3$ dissolved in acidified Milli-Q water)

Incubations were run for ~51.5-55 hours (see Table 1), and at the end of the experiments each polycarbonate bottle was sub-sampled for a FRRf measurement (see below).


## 2.3 Photosynthetic competency: FRRf measurements

Photosystem II (PSII) variable ChlF was measured on a Fast Repetition Rate fluorometer (FRRf, Chelsea Technologies Group FastOcean Sensor fitted with an Act2 laboratory system) using the factory-supplied Act2Run software. All measurements were made in a temperature-controlled room that was kept at the ambient SST temperature (10-14°C). Prior to

each measurement, a blank was prepared by filtering a small aliquot of sample through a 0.2 μm syringe filter, and was subsequently subtracted from all measured fluorescence signals. The excitation wavelength of the FRRf's light-emitting diodes (LEDs) was 450 nm, and the instrument was used in single turnover mode, with a saturation phase comprising 100 flashlets on a 2 μs pitch and a relaxation phase comprising 40 flashlets on a 60 μs pitch. Samples were low-light acclimated for ~1 h prior to each fluorescence light curve measurement to ensure complete relaxation of all NPQ. Measurement of each

fluorescence light curve took 24 min and comprised 8 light steps of increasing intensity, with the majority of light steps lasting 30 seconds except for the following: darkness (300 seconds), the final, highest light step (600 seconds), the previous light step (300 seconds) and the light step before that (120 seconds). These durations were chosen to "gently" induce NPQ at high light. Fluorescence light curves were optimized to yield estimates of NPQ capacity (details below) at light intensities of either 750 μmol quanta $m^{-2}$ $s^{-1}$ (for samples from the underway seawater system) or 1000 μmol quanta $m^{-2}$ $s^{-1}$ (incubation



samples). These high-light intensities were chosen based on experimentation earlier in the voyage and represent a balance such that a curve could still be fitted to the fluorescence induction data, while maximizing the NPQ signal.

Three measured fluorescence signals, $F_o$, $F_m$ and $F_m'$ were used to calculate fluorescence parameters, following Rohacek (2002). $F_o$ and $F_m$ refer to minimum and maximum fluorescence in the dark-acclimated state, while $F_m'$ is the maximum

fluorescence in the light-regulated state, in this case measured at the highest light level of the respective fluorescence light curve.

The maximum quantum yield of PSII, Fv/Fm, was estimated from fluorescence measurements during the dark step of the fluorescence light curve as follows:


$$\frac{F_v}{F_m} = \frac{F_m - F_o}{F_m} \tag{1}$$

There are a number of different NPQ formulations available in the literature (e.g., see Rohacek, 2002), with their respective advantages and disadvantages. We chose to focus on two parameterizations: Stern-Volmer (NPQ$_{SV}$) and the normalized

Stern-Volmer NPQ (NPQ$_{NSV}$):

$$NPQ_{SV} = \frac{F_m - F_m'}{F_m'} \tag{2}$$

$$NPQ_{NSV} = \frac{F_o'}{F_v'} \tag{3}$$


with $F_o'$ calculated following Oxborough and Baker (1997): $\quad F_o' = \frac{F_o}{\left(\frac{F_v}{F_m} + \frac{F_o}{F_m'}\right)} \tag{4}$

and $\qquad F_v' = F_m' - F_o' \tag{5}$

Both parameters were estimated at the respective maximum light levels of the fluorescence light curves and should thus be

regarded as a NPQ "capacity" which can be achieved under given nutrient availabilities and light histories of the phytoplankton assemblage (*c.f.* Schuback and Tortell, 2019). The light levels at which the NPQ parameters were measured are indicated in parentheses in the respective Figures; see also Table 2 for a summary of fluorescence parameters discussed in this paper.

**2.4 Phytoplankton pigments and absorption**

**2.4.1 HPLC analyses**

Sample volumes of 3-4 litres were filtered through a 25 mm glass fibre filter (Whatman GF/F), blotted dry and stored at –80°C until analysis. Samples were extracted over 15–18 hours in an acetone solution before analysis by high performance

liquid chromatography (HPLC) using a $C_8$ column and binary gradient system with an elevated column temperature following the method of Clementson (2013). Pigments were identified by retention time and absorption spectrum from a photo-diode array (PDA) detector, and concentrations of pigments were determined from commercial and international standards (Sigma, USA; DHI, Denmark).

**2.4.2 Diagnostic pigment indices**





The results from HPLC analyses were used to derive diagnostic pigment (DP) indices following Barlow et al. (2008), which give indication of the phytoplankton community composition in terms of three functional groups: diatoms, flagellates and prokaryotes. The DP index and resulting functional groups are defined based on 7 biomarker pigments as follows:

DP = (alloxanthin (Allo) + 19'-hexanoyloxyfucoxanthin (Hex-fuco) + 19'-butanoyloxyfucoxanthin (But-fuco) + fucoxanthin

(Fuco) + zeaxanthin (Zea) + chlorophyll b (Chl-b) + peridinin (Perid))

Diatoms = Fuco/DP

Flagellates = (Allo + Hex-fuco + But-fuco + Chl-b)/DP

Prokaryotes = Zea/DP

### 2.4.3 Fluorometric measurement of chlorophyll

Samples for fluorometric Chl-a analyses (2 L) were filtered onto 25 mm glass fibre filters (Whatman, GF/F) and immediately frozen at –80°C. Within 3 weeks of sampling, all filters were extracted with 10 mL of 90 % acetone at –20°C for 24 hours. Fluorescence was measured on a Turner Trilogy Laboratory Fluorometer and converted to Chl-a using a standard chlorophyll-a dilution that was calibrated spectrophotometrically (Jeffrey and Humphrey, 1975).


Fluorometric Chl-a showed excellent agreement with Chl-a measured using the HPLC method, albeit with an offset and a slight decrease in the slope relative to the 1:1 line (Fig. S1 in SI). In order to bring the Chl-a measured fluorometrically in line with the more precise HPLC measurement (e.g., Trees et al. 1985), all fluorometric Chl-a estimates were corrected using the slope and intercept from the regression of HPLC Chl-a against fluorometric Chl-a (Chl$_{HPLC}$ = Chl$_F$ * 0.879 - 0.052; r$^2$ =

0.97, n = 15, standard error of the estimate = 0.029 mg m$^{-3}$). All Chl-a estimates could thus be combined for calibration of the ac-9 absorption line height approach (see below).

### 2.4.4 Phytoplankton absorption (filter pad):

Sample volumes of 3–4 litres were filtered through a 25 mm glass fibre filter (Whatman GF/F), and the filter was then stored

flat at –80°C until analysis. Optical density spectra for total particulate matter were obtained using a Cintra 404 UV/VIS dual beam spectrophotometer equipped with an integrating sphere. Quartz glass plates were used to hold the sample and blank filters against the integrating sphere. The optical density of the total particulate matter of each sample was obtained using a blank filter as a reference (from the same batch number as the sample filters) wetted with filtered seawater (0.2 μm) and scanned from 200–900 nm with a spectral resolution of 1.3 nm. Following the scan for total particulate matter, the sample

filter was returned to the original filtering units, and any pigmented material was extracted using the method of Kishino et al. (1985). Blank reference filters were treated in the same manner. The filters were rinsed with filtered seawater and then rescanned to determine the optical density of the detrital or non-algal matter. An estimate of the optical density due to phytoplankton was obtained as the difference between the optical density of the total particulate matter and the detrital or non-algal matter. The optical density scans were converted to absorption spectra by first normalizing the scans to zero at 750

nm and then correcting for the path length amplification using the coefficients of Mitchell (1990).

### 2.4.5 Particulate absorption (ac-9):

A WETLabs ac-9 instrument with 25-cm flow tubes was employed in underway mode on the underway seawater line of the *RV Investigator*. The instrument was immersed in the upright position in a flow-through water bath for temperature control,

and the flow tubes were shielded from ambient light. Flow was upwards through the tubes and trickled from there into the water bath, where an overflow valve controlled the water level. About 2/3 of the instrument was immersed, with the upper part exposed to the atmosphere. Internal instrument temperature was monitored every few hours and fluctuated between 21 and 26°C. The inflow was manually switched between unfiltered and filtered seawater (0.8/0.2 μm; AcroPak 1500 capsule




filter with Supor membrane), with filtered measurements lasting ~10 min and conducted every ~2 h. Flow rate was 1–1.5 L
min⁻¹ for unfiltered seawater and not lower than 0.5 L min⁻¹ for filtered seawater. The filter was exchanged for a new one
after 10 days.

Because we were interested only in the absorption line height at 676 nm, only the red absorption channels of the ac-9 (λ =
650, 676 and 715 nm) were processed. Analysis of repeat measurements on Milli-Q water in the laboratory prior to the
voyage (2–7 minutes each, then the average taken for each measurement) indicated that the precision of these 3 channels
ranged 0.0023 – 0.0028 m⁻¹ (2.3–2.9 %) when expressed in terms of the standard deviation of measurements (n=8). The
absolute range, determined as the difference between the maximum and minimum value of the measurements, was 0.0059 –
0.0071 m⁻¹ (5.9–7.4 %).

Bubbles in the flow tubes were a common problem on the voyage and were dealt with by applying a thorough cleaning
routine as well as smoothing of the time series during processing (see SI for details). Particulate absorption was calculated
by subtracting interpolated filtered measurements (i.e., dissolved absorption) from unfiltered measurements (total
absorption) at each wavelength (Slade et al., 2010). The particulate absorption line height at 676 nm (LH(676), m⁻¹) was then
calculated following Roesler and Barnard (2013), subtracting a baseline based on absorption measurements at 650 and 715
nm. This LH(676) was calibrated against discrete Chl-a (Chl-a from fluorometric and HPLC measurements combined, see
above) and also against phytoplankton absorption as determined with the filter pad method (see SI).

### 2.5 Simulated *in situ* measurements of size fractionated primary production

Photosynthetic production of organic matter was measured twice during the voyage (7 March and 18 March, 2018; both
times with water from SST <11.5°C) by the 14-Carbon (¹⁴C) tracer method. Algal carbon fixation was measured on samples
collected pre-dawn from trace metal clean Niskin bottles deployed on a TMR from 3 or 4 depths: 35, 50, (70 m for the 18
March incubation) and 100 m. Sampling depths were determined from *in situ* irradiance depth profiles obtained during
midday CTD casts the day prior to collection. Samples were dispensed into 300 mL acid-washed polycarbonate bottles and
spiked with 16 µCi of Sodium ¹⁴C-bicarbonate (NaH¹⁴CO₃; specific activity 1.85 GBq mmol⁻¹; PerkinElmer). Following the
addition of ¹⁴C, samples were incubated for 24 h in neutral density mesh bags in a deckboard incubator. The temperature of
the incubator was controlled by a continuous supply of surface seawater. Six samples (5 light and one dark bottle per
irradiance) were incubated under natural sunlight at 6 light intensities (from 67 to 0.2 % of incident irradiance), which were
adjusted by varying the layers of neutral density mesh. Light attenuation was measured with a Biospherical Instruments
QSL2101 Quantum Scalar PAR Sensor. Carbon fixation based on radioisotope measurements and 24-hour incubations are
reported to approximate net primary production (Laws, 1991).

Upon completion of the 24-h incubation, 4 replicate samples were filtered in series through 0.2, 2.0, and 20 µm
polycarbonate filters (Poretics) separated by 200 µm nylon mesh, and two samples were filtered through 0.2 µm filters (a
total community 'light' control, and a total community 'dark' control). Data for the dark-corrected size-fractionated samples
are reported here. Filters were rinsed with 0.2 µm-filtered seawater, acidified to volatilize any remaining inorganic carbon
(Boyd and Harrison, 1999), and collected in 20 mL glass scintillation vials (Wheaton) to which 10 mL of liquid scintillation
cocktail (UltimaGold, PerkinElmer) was added. Samples were subsequently analysed by liquid scintillation counting
(PerkinElmer Tri-Carb 2910 TR). Water column integrated carbon assimilation rates were calculated using the trapezoid rule
with the shallowest value extended to 0 m and the deepest extrapolated to a value of zero at 200 m.


### 2.6 Mooring data



We show SOTS mooring data from the Pulse-7 deployment in 2010-2011. The mooring was equipped with a number of instruments including a downward-looking WETLabs ECO FLNTUS fluorometer with a "bio-wiper" at 30 m and a wiped ECO PAR (photosynthetically available radiation) sensor at the same depth. Seawater temperature was measured at 15

discrete depths with Vemco Minilog Classic sensors (except for the shallowest sensor at 30 m, which was a SeaBird Electronics SBE16plusV2), and mixed layer depth (MLD) was calculated based on a temperature difference of 0.3°C relative to the 30 m measurement. Only data with a quality flag <2 were used, and the fluorescence sensor had been calibrated using factory-supplied dark values and scale factors (Schallenberg et al., 2019). Fluorescence-based WETLabs ECO sensors have been shown to overestimate Chl-a in the Southern Ocean by up to a factor 7 compared to pigment

samples, with factor ~4 overestimation reported for the South Indian Ocean (Roesler et al., 2017). We have thus divided the fluorescence output from the FLNTUS sensor by 4 when Chl-a concentration as a proxy of phytoplankton biomass was the entity of interest, consistent with previous approaches to fluorescence data from the SAZ (e.g., Eriksen et al., 2018). However, the fluorescence data were also used to estimate NPQ over the daily cycle, and for this exercise fluorescence was not divided by 4, nor was it in any other way normalised to biomass.


NPQ from the FLNTUS fluorometer was estimated analogously to the Stern-Volmer parameterization

$$NPQ_{SF} = \frac{F_{max} - F_{min}}{F_{min}} \qquad (6)$$

where $NPQ_{SF}$ stands for NPQ measured with a standard fluorometer and $F_{max}$ and $F_{min}$ are the maximum and minimum fluorescence measurements made with said fluorometer in a day, with the following restrictions: the daily fluorescence and

PAR data were first smoothed with a Loess filter, and $NPQ_{SF}$ was only estimated if the minimum fluorescence was within 2 hours of maximum PAR. The parameter $F_{max}$ designates the maximum smoothed fluorescence between 5 am and noon on a given day. This was preferable to using $F_{max}$ from night-time because it provides that the two fluorescence measurements are closer in time, and thus the likelihood of $F_{max}$ varying due to fluctuations in Chl-a concentrations is reduced. Furthermore, this approach corresponds to analyses carried out on the underway fluorescence data from the SOTS voyage, discussed

below. Note that $NPQ_{SF}$ is a "realized" NPQ at incident light, in contrast to the NPQ "capacity" estimated with the FRRf.

## 3. Results and Discussion

### 3.1 Incubation experiments: sensitivity of ChlF parameters to Fe status

The incubation experiments were designed with the goals to: 1) test whether the resident phytoplankton were Fe-limited and 2) investigate how NPQ capacity and Fv/Fm, as derived from FRRf measurements, respond to changes in iron status. All 3 incubations were carried out with source waters colder than 11.5°C (Table 1); i.e., with waters from the colder water mass as evident in Fig. 1, and the respective pigment compositions of the resident phytoplankton were very similar (Fig. S2), indicating a dominance of haptophytes, with some diatoms and chrysophytes also present. The pooled data from the

experiments show unequivocally that Fe addition increased the maximum quantum yield of PSII, Fv/Fm, as has been observed previously in HNLC regions (e.g., Boyd and Abraham, 2001; Kolber et al., 1994). The NPQ capacity, measured as both $NPQ_{SV}(1000)$ and $NPQ_{NSV}(1000)$, decreased with iron addition (Fig. 2). The functional absorption cross section of PSII, $\sigma_{PSII}$, also decreased with Fe addition, as has been observed previously (Kolber et al., 1994; Schuback et al., 2015; Schuback and Tortell, 2019). Overall, $\sigma_{PSII}$ were very large in all treatments (~10–12 $nm^2$ $RC^{-1}$), as has been observed previously for

low Fe-adapted Southern Ocean phytoplankton (Strzepek et al. 2012; Strzepek et al., 2019). The addition of DFB, a strong organic ligand (siderophore) that decreases Fe availability to phytoplankton, had the opposite effect to Fe addition for all parameters. The phytoplankton thus showed a clear response to changes in Fe status, which affected their ability to process

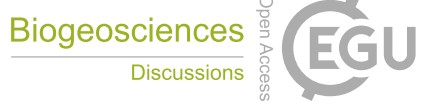

absorbed light energy, evident in a lower maximum quantum yield of PSII, Fv/Fm, and an increased capacity and need to dissipate excess excitation energy as NPQ.


For all parameters except $\sigma_{PSII}$, the results for the +Fe treatment were significantly different from the respective control (Wilcoxon rank sum test, p<0.05), indicating that phytoplankton in the sampled water mass, i.e. the cold waters <11.5°C, were Fe-limited. This view is further supported by the low initial Fv/Fm observed for all experiments (~0.33, Table 1); a fully dark-regulated Fv/Fm in this range is a widely accepted indicator of Fe limitation in HNLC waters (Boyd and

Abraham, 2001; Kolber et al., 1994; Suggett et al., 2009).

The changes in NPQ capacity were diametrically opposite to the changes in Fv/Fm with respect to Fe status: as Fe stress increased, so did the capacity for NPQ. Similar NPQ responses to Fe status have been observed previously in controlled experiments both in the laboratory and with natural phytoplankton communities from Fe-limited ocean regions (e.g.,

Alderkamp et al., 2012; Petrou et al., 2014; Schuback et al., 2015; Schuback and Tortell, 2019). While some of these studies reported $NPQ_{SV}$, others estimated $NPQ_{NSV}$, hence we decided to investigate the Fe-effect on both. Non-photochemical quenching affects all fluorescence levels and is easiest quantified using the Stern-Volmer parameterization, which was originally developed for higher plants. However, this parameterization does not take into account differences in Fv/Fm in the dark-regulated state, which can mask differences in NPQ as only the light-induced component is assessed, ignoring

differences in basal levels of heat dissipation. The Normalized Stern-Volmer parameterization explicitly accounts for differences in Fv/Fm, which means it is a more appropriate parameter when samples with different Fv/Fm are being compared, as is the case in our incubations with different levels of Fe stress. However, it is important to note that mathematically, $NPQ_{NSV}$ is inversely related to Fv/Fm when $F_o$' is calculated (see Eq 4) rather than measured directly, as is the case in our study:

$$NPQ_{NSV} = \frac{\frac{F_o}{F_m'}}{\frac{F_v}{F_m}} \qquad (7)$$

This inverse mathematical relationship, in conjunction with the established sensitivity of Fv/Fm to Fe limitation (Boyd and Abraham, 2001; Kolber et al., 1994; Suggett et al., 2009), means that $NPQ_{NSV}$ is not a truly independent parameter with respect to Fe status, even though the inverse relationship with Fv/Fm may be physiologically relevant. Furthermore, measurement of $NPQ_{NSV}$ requires knowledge or calculation of $F_o$ or $F_o$' (Eq. 3-5), which means it can only be measured with

an active fluorometer, such as a FRRf. Conversely, parameters analogous to the $NPQ_{SV}$ parameter can be estimated with any fluorometer so long as a dark-regulated and light-regulated measurement is available.

A physiological relationship between NPQ and Fe stress can be understood conceptually as a result of increased excitation pressure on the reaction centres when Fe is limiting (Schuback et al., 2015). Under Fe stress, there are fewer reaction centres

and electron transport chains per chlorophyll because electron transport chains are Fe-expensive (Behrenfeld and Milligan, 2013). More energy is thus funnelled to fewer reaction centres, which causes the increase in $\sigma_{PSII}$ (Alderkamp et al., 2019; Ryan-Keogh et al., 2017; Strzepek et al., 2012). This arrangement comes at the expense of the ability to deal with fluctuations in light; the system is "less robust". Iron limitation thus increases the need for rapid photoprotection, which can be achieved through an increased capacity for NPQ, as has been observed previously (Schuback et al., 2015, Schuback and

Tortell, 2019).

From the data presented above, we conclude that the resident phytoplankton in the colder water mass were indeed Fe-limited (evident in the clear treatment response to Fe-addition, Fig. 2), and that both the NPQ capacity and Fv/Fm, as derived from FRRf measurements, responded to changes in Fe status. While Fv/Fm increased with Fe addition, the NPQ capacity





decreased, consistent with expectations. The observation that the colder water mass exhibited signs of Fe-limitation is also
consistent with expectations, as the SAZ, and in particular the SOTS site, has been shown to be seasonally Fe-limited, with
limitation increasing towards the end of summer (Boyd et al., 2001; Hutchins et al., 2001; Lannuzel et al., 2011; Sedwick et
al., 1999).

**3.2 Underway data: two SST regimes with different ChlF parameters (FRRf)**

Fluorescence measurements similar to the ones carried out on the incubations were also performed on samples taken from
the underway seawater line on the SOTS voyage (n=66). Fv/Fm ranged from 0.2 to 0.5 in the resident phytoplankton and
showed an inverse relationship with NPQ capacity (Fig. 3), as was observed for the incubations. The trends in ChlF
parameters appeared to fall into two groups based on sea surface temperature (SST): phytoplankton in the colder water mass
(<11.5°C) showed lower Fv/Fm and higher NPQ capacity, consistent with increased Fe limitation, compared to the
phytoplankton in the warmer water mass (>13.5°C). Grouping the ChlF parameters based on SST illustrates this point even
further (Fig. 4): Fv/Fm, as well as the capacity to dissipate excess excitation energy (estimated as $NPQ_{NSV}$ and $NPQ_{SV}$), were
significantly different between the two water masses ($p \ll 0.01$, Wilcoxon rank sum test). It thus appears that the
phytoplankton in the two SST regimes experienced different levels of Fe stress, but before such a conclusion can be drawn, a
number of confounding factors must be discussed, including species composition, mixed layer depth, and the effect of
temperature on physiology.

**3.2.1 Phytoplankton pigments and community composition in the two SST regimes**

The diagnostic pigment (DP) index, calculated from HPLC pigment analyses, provides a simple though not definitive means
to investigate phytoplankton community composition. It focusses on three major groups (diatoms, flagellates and
cyanobacteria) selected on the basis of their significance rather than their respective size ranges (Barlow et al., 2008). Such a
diagnostic index is naturally a simplification and should be treated with caution. However, the DP index has been used
across a number of ecological ocean regions, including the SAZ and Southern Ocean, where it compared favourably to the
more elaborate CHEMTAX analyses (e.g., Mendes et al., 2015).


Our analysis indicates a dominance of flagellates in both the warm and cold SST regimes, with diatoms and prokaryotes
playing a lesser role (Fig. 5). Despite some differences between the two SST regimes, the overall pattern is similar,
indicating that the phytoplankton assemblages in the two water masses were comparable (see Fig. S3 and S4 for detailed
pigment and phytoplankton absorption results). Moreover, these distributions among major phytoplankton groups are
consistent with the phytoplankton community composition in the SAZ reported by Mendes et al. (2015) using similar
methods. These authors also conducted a CHEMTAX analysis of their data that suggested a dominance of haptophytes in the
SAZ, with pelagophytes, prasinophytes and diatoms making up the majority of other taxa. A recent microscopic analysis of
fortnightly samples from a remote sampler on the SOTS mooring at 142°E, 47°S likewise indicated that flagellates were the
most abundant phytoplankton group between September and April, followed by diatoms (Eriksen et al., 2018). The results
from the DP index analysis are thus consistent with previous estimates for the region and show that the two SST regimes
were broadly similar with respect to phytoplankton community composition.

**3.2.2 Mixed layer depths**

The depth of the mixed layer can have a profound impact on NPQ capacity, as phytoplankton acclimate to their light
exposure. Deeper mixed layers, resulting in stronger fluctuations in light availability, have been found to increase the NPQ
capacity of phytoplankton in the field and under simulated conditions in the laboratory (Browning et al., 2014; Milligan et
al., 2012). In our study region, the mixed layer depth was deepest in the warm SST regime (~100 m; Fig. S5), while it ranged





between 20 and 90 m in the colder regime. If mixed layer depth were the driver for the observed differences in NPQ capacity
between the two water masses, we would expect that the warmer water mass show increased NPQ capacity. However, the

observed NPQ trend is exactly opposite, with higher NPQ capacity evident in the colder water mass. With only one CTD
station in the warmer SST regime, we may have under-sampled this water mass, but it is unlikely that it showed overall
shallower mixed layers than the cold-water regime. We thus conclude that the observed trend in NPQ capacity can not be
explained by the mixed layer depths encountered.

### 3.2.3 Temperature effects on phytoplankton photophysiology

It is well established that temperature has a strong effect on phytoplankton growth, especially with respect to the optimal
niche for a given species (e.g., Boyd et al., 2013; Davison, 1991; Raven and Geider, 1988). However, the maximum
quantum yield of PSII, Fv/Fm, appears less sensitive to temperature. Manipulative studies have reported little or no effect on
Fv/Fm for phytoplankton grown at temperatures differing by 4–8°C between minimum and maximum temperature (e.g.,

Kulk et al., 2012; Rose et al., 2009). Indeed, the latter study specifically investigated the synergistic effects of Fe and
temperature on Antarctic phytoplankton. An increase in temperature by 4°C did not result in any change of Fv/Fm compared
to the control, but an addition of Fe increased Fv/Fm significantly – with and without an increase in temperature (Rose et al.,
2009). We thus conclude that it is highly likely that the differences in Fv/Fm and NPQ capacity we observed between the
two SST regimes (with mean SSTs of 10.8±0.5°C and 14.6±0.7°C, respectively), were the result of physiological changes in

the phytoplankton due to their nutritional (Fe) status, rather than being caused by the different ambient temperatures.

### 3.2.4 The case for differences in physiological status in the two SST regimes

The strongest indicator that the two SST regimes held phytoplankton communities with different levels of Fe stress is the
difference in Fv/Fm between the two regimes (Fig. 4). The decrease in Fv/Fm in the colder water mass corresponded to a

large increase in $F_o$ relative to the warm regime (Fig. S6), in line with the hypothesis that under Fe limitation damaged
and/or disconnected light harvesting complexes contribute to background fluorescence, as has been previously observed
(Behrenfeld and Milligan, 2013). The corresponding increase in $F_m$ in the presumably Fe-limited water mass is also not
unexpected and has been observed in other Fe-limited regions (e.g., Behrenfeld and Kolber, 1999).

Rates of water column integrated net primary productivity measured in the cold SST regime also point to Fe limitation in
that water mass. Column integrated primary production rates ranged from 317±30 mg C m$^{-2}$ d$^{-1}$ (18 March 2018) to 500±104
mg C m$^{-2}$ d$^{-1}$ (7 March 2018). These rates agree well with those measured at the SOTS site in March 2016 using the same
method (670±25 mg C m$^{-2}$ d$^{-1}$) (Ellwood et al., submitted), but are considerably lower than those measured by Westwood et
al. (2011) in subantarctic waters in austral summer (January-February 2007) when Fe concentrations are higher: 1034-1627

mg C m$^{-2}$ d$^{-1}$. While part of the difference between our results and those of Westwood et al. (2011) may be due to
methodological differences (i.e., short (2h) versus long (24h) $^{14}$C incubations), we ascribe most of the difference to the
seasonal cycle of Fe limitation observed previously for SOTS (Boyd et al., 2001; Hutchins et al., 2001; Lannuzel et al.,
2011; Sedwick et al., 1999).

Fewer ancillary data are available for the warmer water mass, as there were no primary productivity measurements or Fe-
addition experiments undertaken. However, macronutrient data from one CTD in that water mass indicate that the warm SST
regime was not High Nutrient, Low Chlorophyll (HNLC), as NOx was drawn down to near-zero, which was not the case for
the colder waters where NOx was always well above 5 μmol L$^{-1}$ (Fig. S7). Phosphate concentrations were also significantly
lower in the warmer waters than the cold waters, while silicate was depleted for both.




The SST map in Fig. 1 indicates that the warm SST signature near SOTS was part of a cyclonic eddy, which could be tracked back in time to even warmer waters south-west of Tasmania based on its signature on satellite altimetry and SST images (not shown). At the time of sampling, the eddy-associated water mass exhibited SSTs reaching >14°C, compared to <12°C for the waters further west, and it showed warmer and saltier water throughout the water column, with a salinity

minimum around 34.4 (Fig. S8). Such a T-S signature is consistent with an origin of the eddy-associated water mass either east or west of Tasmania, with a strong influence from the Tasman Sea (Herraiz-Borreguero and Rintoul, 2011). Waters east of Tasmania have previously been found to be Fe-replete (and low in nitrate), with airborne dust and shelf sediments presumed to be the main Fe sources (Bowie et al., 2009; Lannuzel et al., 2011). $F_v/F_m$ values indicative of Fe sufficiency (~0.5) were also reported for the SAZ south-east of Tasmania (SST ~15°C) in late summer, along with very low nitrate and

non-limiting Fe concentrations at the surface (Hassler et al., 2014). It is thus highly likely that the warmer water mass encountered during the SOTS voyage in 2018 was Fe-replete, while the colder water mass was Fe-limited. The absence of dissolved Fe data leaves the Fe nutritional status of the warmer water mass community less than completely clear (although the presence of $F_v/F_m$ values above 0.4 suggests Fe limitation was unlikely since Fe status is known to be the dominant driver for $F_v/F_m$ in the Southern Ocean, e.g., Suggett et al. (2009)). Regardless of this uncertainty, the Fe-limited status and

corresponding high NPQ capacity of the cold-water community is clear.

### 3.3 Underway data: Continuous measurements across the two SST regimes

The two SST regimes were repeatedly visited by the ship during the voyage (Fig. 6). The Chl-a data from the ac-9 were remarkably different from the fluorometer traces (compare Fig. 6b) and 6c)), with clear day-night fluctuations in the latter

that were not reflected in the ac-9 data. These fluctuations were especially pronounced in the colder water mass, which had overall lower Chl-a concentrations than the warmer waters (mean Chl-a = 0.38±0.06 mg m$^{-3}$ for SST<11.5°C and 0.62±0.11 mg m$^{-3}$ for SST>13.5°C). The daily fluorescence minima coincided with the maxima in above-water incident photosynthetically available radiation (PAR) and vice versa: i.e., fluorescence was highest during the night. Such daily fluorescence cycles are consistent with non-photochemical quenching of fluorescence in the daytime.


### 3.3.1 Physiological information in underway fluorescence: NPQ differences between the two SST regimes

In order to investigate NPQ as measured with the standard fluorometer (Fig. 6c) in more detail, we normalized fluorescence by Chl-a as estimated by the ac-9 and plotted it against incident daytime PAR (Fig. 7). Colour-coding by SST clearly shows that two NPQ regimes were at play. In the colder waters, F/Chl-a was high at low PAR and decreased considerably as PAR

increased – dynamic NPQ was high. In the warmer waters, F/Chl-a was relatively constant across PAR values and dynamic NPQ was thus low. Note that normalization to Chl-a removes differences in the fluorescence signal that would be caused by differing Chl-a concentrations in the water masses.

The NPQ signal from the standard fluorometer was further investigated by formally separating the data from the two water

masses based on SST (Fig. 8). A Loess filter was applied to the respective pooled data sets (for values measured at PAR>5 and PAR<1000 µmol quanta m$^{-2}$ s$^{-1}$), and NPQ$_{SF}$ ("NPQ measured with a standard fluorometer") was estimated from the filtered data according to Eq (6). The NPQ$_{SF}$ signal in the colder, HNLC water mass was more than twice as high as that in the warm SST regime, and a similar trend was discernible even without normalization of the fluorescence signal to Chl-a. The trends in NPQ as evidenced in the data from the standard fluorometer are thus consistent with the NPQ capacities

measured with the FRRf (Fig. 4): increased dynamic NPQ (and NPQ capacity) in the colder waters relative to the warm SST regime. This result is significant for it indicates that measurements from a standard fluorometer can be interpreted with regards to phytoplankton physiological status. Without the control afforded by an FRRf, simply with a standard fluorometer and the daily cycle of the sun, a signal is measured (NPQ$_{SF}$) that holds profound physiological information. This shouldn't





come as a complete surprise, as it has long been recognized that standard fluorometers are affected by NPQ, and
considerable research has gone into investigating how to best correct for NPQ in order to retrieve more accurate profiles of
Chl-a concentration (Biermann et al., 2015; Thomalla et al., 2018; Xing et al., 2012, 2018). However, given the dependence
of NPQ on incident irradiance, attempts to correct for it in a "physiological" way have tended to be based on irradiance
thresholds or light history (e.g., Behrenfeld et al., 2009; Xing et al., 2018), while ignoring other factors affecting
phytoplankton physiological status, such as Fe limitation.


A noteworthy exemption is the study by Browning et al. (2014), which specifically investigated the variability of NPQ along
eco-physiological gradients in the Southern Ocean. That study showed similar trends to ours: increased dynamic NPQ was
associated with Fe-limited waters of the Antarctic Circumpolar Current, while subtropical gyre-type waters, where Fe was
replete and nitrogen was low, exhibited subdued NPQ. Overall, they found that their NPQ parameterization showed the

strongest correlation with SST, which they interpreted to be largely caused by the relationship between SST and the extent of
near-surface stratification and mixed layer depth (Browning et al., 2014). They also found a correlation between NPQ and
Fv/Fm that was statistically significant at the $p<0.001$ confidence level, but was weaker than that with SST. It is worth
pointing out that there were significant methodological differences between their approach and ours, as well as a much larger
SST gradient in their study region (3-25°C), which likely has its own correlation with Fe limitation while also causing

differences in species composition that can have a strong influence on Fv/Fm (Suggett et al., 2009). Regardless of the cause
in NPQ variability, our findings reinforce the observation by Browning et al. (2014) that NPQ can be highly variable due to
factors other than incident PAR, implying that NPQ corrections that rely solely on PAR should be viewed with caution.

Overall, our results are consistent with the differences in NPQ observed in the two SST regimes being caused by differences
in their Fe limitation status, as evidenced in the respective Fv/Fm, with implications for the expected efficiency of
photochemistry. Not only did we find a strong inverse relationship between Fv/Fm and NPQ capacities estimated with an
FRRf light-curve protocol, but the relationship was also borne out in $NPQ_{SF}$ measured with a standard fluorometer – an
instrument type frequently used on moorings, ships and autonomous measuring platforms such as Argo floats. We have thus
shown that measurements from a standard fluorometer can provide useful information on phytoplankton physiology, in
particular the presence of Fe stress.

### 3.4. Case study: Application of $NPQ_{SF}$ approach to mooring data

In order to test further the utility of this interpretation of fluorescence measured with a standard fluorometer, we investigated
$NPQ_{SF}$ on a timeseries from the SOTS mooring (Fig. 9a). Here, $NPQ_{SF}$ was estimated based on daily fluorescence
measurements, after application of a smoothing filter (see Section 3.3.1). Due to differences in the maximum PAR on any
given day, one could argue that $NPQ_{SF}$ should be normalized by the incident PAR at $F_{min}$, as the depression in fluorescence is
expected to be proportional to the incident light. Such a measure decreases the scatter in the time series but preserves the
overall trend observed in Fig. 9a) (Fig. S9). Note that noise in $NPQ_{SF}$ is likely related to both fluctuating light conditions day
to day and patchiness in phytoplankton fields. The fluorometer on the SOTS mooring measured once every hour, so only a
very limited number of data points were available for any given day. The passages of small fronts and phytoplankton patches
would thus have a large impact on the trends observed in a day, confounding the NPQ signal.

In the context of the ancillary measurements (Fig. 9), the trends observed in $NPQ_{SF}$ over the growing season appear sensible.
Some connection with changes in the MLD is apparent, for example a decrease in $NPQ_{SF}$ in November as the mixed layer
shoals, but the MLD dynamics cannot explain fluctuations in $NPQ_{SF}$ over the summer. Nor does the light field experienced
within the mixed layer (Fig. 9c) show a clear relationship with $NPQ_{SF}$. However, there is an interesting connection with the





dynamics in Chl-a concentrations: $NPQ_{SF}$ is relatively low in spring and remains low, with some fluctuations, until Chl-a peaks in mid-January. After this peak, Chl-a decreases and $NPQ_{SF}$ increases, a trend that could indicate the onset of Fe-limitation at the SOTS mooring late in summer. Indeed, several studies have concluded that the SAZ, in particular the SOTS

area, is Fe-limited in summer (Boyd et al., 2001; Hutchins et al., 2001; Lannuzel et al., 2011; Sedwick et al., 1999), with Fe re-supply in winter from Ekman transport (Ellwood et al., 2008) and deep mixing (Tagliabue et al., 2014). An interpretation of increased $NPQ_{SF}$ as a response to the onset of Fe limitation is thus supported by the historical data available. Furthermore, we point out that the increase in $NPQ_{SF}$ corresponds to an increase in water temperature over the season – the trend is thus opposite to the one seen in the voyage data, where $NPQ_{SF}$ was highest in the colder waters. This further supports our

contention that the observed trends in $NPQ_{SF}$ (and Fv/Fm) are not driven by SST changes.

This same time period on the SOTS mooring has also been evaluated based on discrete samples for nutrients and phytoplankton, collected by an autonomous sampler (Eriksen et al., 2018). Of the macronutrients, only silicate decreased to concentrations that could be considered limiting (<1 μM at the end of January 2011). This development coincided with a

decrease in diatom biovolume relative to other genera that lasted for about a month, but began to recover in March 2011 (Eriksen et al., 2018). Overall, the phytoplankton assemblage at SOTS in 2010/2011 was very diverse with diatoms, dinoflagellates and ciliates figuring most prominently with respect to biovolume. Similarity-tests based on abundance indicated a spring community that was present until the end of 2010, followed by a summer community until the end of February, and a fall community thereafter (Eriksen et al., 2018). The observed fluctuations in $NPQ_{SF}$ can thus not be tied to

distinct changes in phytoplankton community.

To sum up, the findings from the SOTS mooring for the 2010/2011 season support our argument that $NPQ_{SF}$ may hold information on phytoplankton physiology, as the observed fluctuations in $NPQ_{SF}$ cannot be explained by developments in seasonal MLD, SST, light experienced in the mixed layer, or phytoplankton community composition. Rather, they exhibit a

trend that is consistent with historical studies in the SAZ that have observed Fe limitation developing in summer and into autumn (see above), which could lead to an increased need for photoprotection, as evidenced in higher $NPQ_{SF}$.

## 4. Conclusions and future work

This study has explored the connection between NPQ and phytoplankton physiological status as indicated by Fv/Fm in an

HNLC region. The results suggest that the variability observed in NPQ – mirroring changes in Fv/Fm – was driven by different levels of Fe limitation. A range of NPQ parameters was assessed, measured with both an active fluorometer (FRRf) and a standard fluorometer mounted on the underway seawater supply of a research vessel. The relationship between NPQ and Fe limitation was tested with incubation experiments, where Fe concentrations were controlled by adding Fe and a strong Fe-binding ligand. All experiments showed a statistically significant relationship between Fe status and NPQ capacity

as measured with an FRRf (Fig. 2), as has been observed previously (Schuback et al., 2015, Schuback and Tortell, 2019). Our analysis is novel in that it uses a standard fluorometer (such as is conventionally used to monitor Chl-a concentrations) to derive the NPQ parameter $NPQ_{SF}$, linking its variability to phytoplankton physiology, i.e. Fe limitation status. Furthermore, application of the proposed methodology to data from a standard fluorometer on the SOTS mooring shows that the derived $NPQ_{SF}$ exhibits a pattern that is consistent with the expected seasonal development of Fe limitation at the site.

The approach may thus be promising for the interpretation of standard fluorometer data from HNLC regions, allowing assessment of the relative changes in Fe limitation status along voyage transects and mooring timeseries. Interpretation of fluorescence data in this way will be improved by independent assessments of Chl-a concentration, such as can be achieved with an ac-9. Normalization by Chl-a (or phytoplankton absorption) removes the influence of Chl-a concentration on the fluorescence signal, thus refining the NPQ signal. Overall, our results point to a novel approach for the interpretation of





widely available fluorescence sensor data to assess the role of Fe sufficiency in phytoplankton physiology, and thus productivity, in the Southern Ocean.

Ultimately, this approach to fluorescence has the potential to be applied to data from autonomous floats in the Southern Ocean, such as the BGC-Argo fleet. A recent application to glider data from the SAZ in the Atlantic Ocean has allowed

identification of regions and time periods where Fe limitation was relieved (Ryan-Keogh and Thomalla, in prep). Studies of this kind will improve our understanding of biogeochemical cycles in the understudied Southern Ocean. However, care will undoubtedly be needed in the interpretation of large spatial variations (e.g., when crossing fronts) owing to differences in species composition and mixing regimes.

Future work should include laboratory experiments under fluctuating light conditions, i.e. mimicking mixing in the ocean, to further probe the interplay of light history and Fe status on NPQ and how both affect phytoplankton growth. Care should be taken to not only focus on steady state scenarios, but to also include perturbation experiments, as relaxation of Fe limitation in the Southern Ocean may be a sporadic event, for example if storm-induced deep mixing or dust deposition are the means of Fe fertilization. Overall, assessment of the level of Fe limitation is less interesting than the question how this limitation

affects primary production and growth rate, as these are the quantities that will be felt throughout the ecosystem. The eventual goal would thus be to establish a link between a readily measurable proxy, such as $NPQ_{SF}$, and phytoplankton growth. This could be achieved by further investigations into the relationship between NPQ and the maximum quantum yield of photosynthesis, which has been shown to respond to Fe fertilization (Alderkamp et al., 2009; Hiscock et al., 2008) and has also been incorporated into a new global model of net primary production, albeit currently as a function of light

acclimation only (Silsbe et al., 2016).

**Data availability**

All voyage data (except FRRf and ac-9) are freely available from the CSIRO Data Trawler:
https://www.cmar.csiro.au/data/trawler/survey_details.cfm?survey=IN2018_V02. The mooring data are freely available via

the Australian Ocean Data Network portal: https://portal.aodn.org.au. FRRf and ac-9 data are available directly from the corresponding author and can be requested by email: christina.schallenberg@utas.edu.au.

**Author contributions**

CS conceived the idea for the study and designed and carried out the experimental work on the voyage except for [14]C uptake

experiments, which were done by RFS. CS processed and analysed all data post voyage with the following exceptions: phytoplankton pigment and filter pad absorption measurements were undertaken and interpreted by LAC, and the results of [14]C uptake experiments were analysed by RFS. TT leads the Southern Ocean Times Series facility which provided the moored sensor observations. The details of the inquiry were significantly improved through discussions with RFS, NS, TWT and PWB. CS wrote the majority of the manuscript, with refinements and additions provided by all co-authors.


**Acknowledgements**

We thank the captains and crews of *RV Investigator* and *RV Southern Surveyor* and the Marine National Facility for their support. Special thanks to Julie Janssens for manning the ac-9 during night shift, and to Peter Jansen for help with the mooring data. Diana Davies, and the CSIRO Moored Sensor Systems team were indispensable for the mooring and voyage

field programs, and we are grateful for all their hard work and good cheer. Finally, we thank Peter Hughes and Julie Janssens for nutrient analyses and hydrochemistry support.





**Competing interests**

The authors declare that they have no conflict of interest.


**Financial support**

CS was supported by a Canadian National Sciences and Engineering Research Council (NSERC) postdoctoral fellowship
(PDF-502793-2017) and by the Antarctic Climate and Ecosystems Cooperative Research Centre (ACE CRC) at the
University of Tasmania. The Southern Ocean Time Series autonomous moored observatory is a facility of the Australian

Integrated Marine Observing System, and also received support from the ACE CRC, the Australian Antarctic Science
Program and the Australian Marine National Facility. RFS and PWB received support from the ACE CRC and the Antarctic
Gateway Partnership (Australian Research Council).

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



Table 1. Details of incubation experiments

| Experiment no. | Date (UTC) | Sampling Time (UTC) | Sampled how | SST (°C) | Light level (% of surface) | Duration (h) | Starting Chl-a concentration (mg m$^{-3}$) | Initial Fv/Fm |
|---|---|---|---|---|---|---|---|---|
| 1 | 6-3-2018 | 20:30 | TMR (35 m) | 10.7 | 67 | 53 | 0.35 | 0.33 |
| 2 | 8-3-2018 | 18:40 | FISH | 11.2 | 67 | 55 | 0.30 | 0.32 |
| 3 | 13-3-2018 | 22:30 | FISH | 10.5 | 25 | 51.5 | 0.37 | 0.34 |






Table 2. List of fluorescence-related parameters derived and discussed

| | Parameter | Units | Method |
|---|---|---|---|
| $\sigma_{PSII}$ | Functional absorption cross section of PSII | $nm^2\ RC^{-1}$ | FRRf ST protocol during dark-regulated state |
| F | Fluorescence | Arbitrary | Measured with a standard fluorometer with minimal or no dark-acclimation period |
| $F_{min}$ | Minimum fluorescence in the day (at high light) | Arbitrary | Measured with a standard fluorometer, with minimal or no dark-acclimation |
| $F_{max}$ | Maximum fluorescence in the day (at low light) | Arbitrary | Measured with an standard fluorometer, with minimal or no dark-acclimation |
| Fv/Fm | Maximum quantum yield of PSII | No units | FRRf ST protocol during dark-regulated state; $(F_m - F_o)/F_m$ |
| NPQ | Non-photochemical quenching | No units | Term comprising a number of non-photochemical quenching processes that decrease fluorescence at supersaturating light intensities (e.g. Fm' relative to Fm, and Fmin relative to Fmax) |
| $NPQ_{NSV}(750)$, $NPQ_{NSV}(1000)$ | Normalized Stern-Volmer NPQ at light intensities of 750 and 1000 µmol quanta $m^{-2}\ s^{-1}$ | No units | FRRf ST protocol during light-regulated state; Eq (3) |
| $NPQ_{SV}(750)$, $NPQ_{SV}(1000)$ | Stern-Volmer NPQ at light intensities of 750 and 1000 µmol quanta $m^{-2}\ s^{-1}$ | No units | FRRf ST protocol during light-regulated state; Eq (2) |
| $NPQ_{SF}$ | NPQ analogous to Stern-Volmer NPQ | No units | Measured with a standard fluorometer with minimal or no dark-acclimation; calculated as $(F_{max} - F_{min})/F_{min}$ |





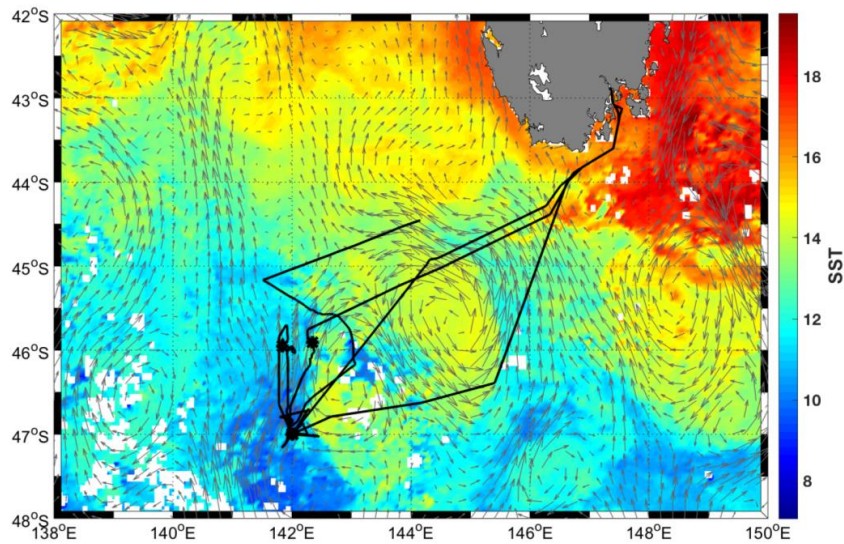


**Figure 1: Average Sea Surface Temperature (SST) for March 2018 from the MODIS-Aqua satellite, overlain with geostrophic currents for March 16, 2018 (grey arrows) and the cruise track (black line) as well as CTD locations (black asterisks). Surface geostrophic velocities (IMOS-OceanCurrent-Gridded sea level anomaly-Delayed mode) were downloaded from the AODN data portal at https://portal.aodn.org.au. Credit: IMOS and CSIRO. References: http://oceancurrent.imos.org.au/**






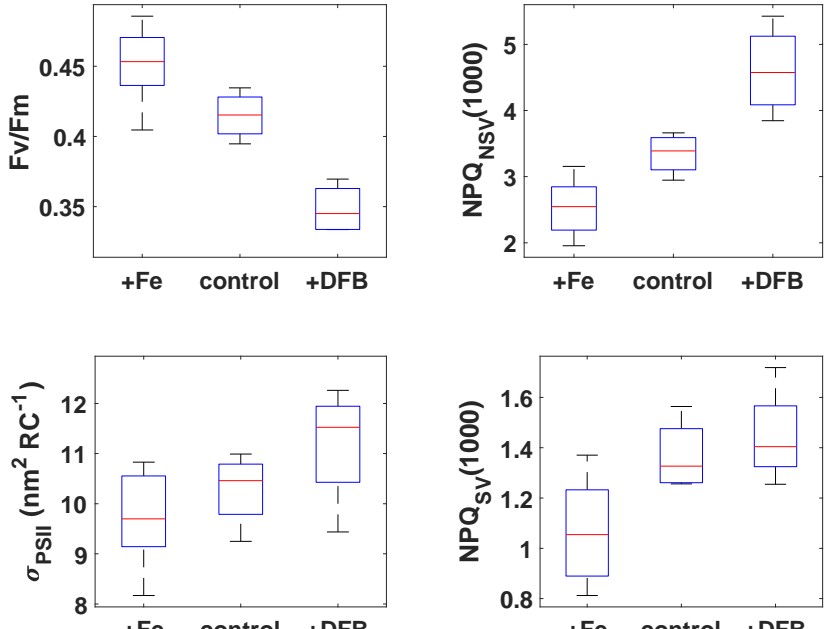

**Figure 2: FRRf results after ~53 hours for the 3 pooled incubation experiments. The respective treatments are indicated on the x-axes (2 nM and 0.2 nM Fe additions were pooled into +Fe), and measured parameters are on the y-axes. NPQ capacity was measured at 1000 µmol quanta m$^{-2}$ s$^{-1}$. Boxplots show the sample medians in red, with the blue boxes indicating the 25$^{th}$ and 75$^{th}$ percentiles; whiskers extend to the most extreme data points. All +Fe treatments were significantly different from the respective DFB treatments (Wilcoxon rank sum test, p<0.05), and except for $\sigma_{PSII}$, all +Fe treatments were also significantly different from the respective control treatments.**



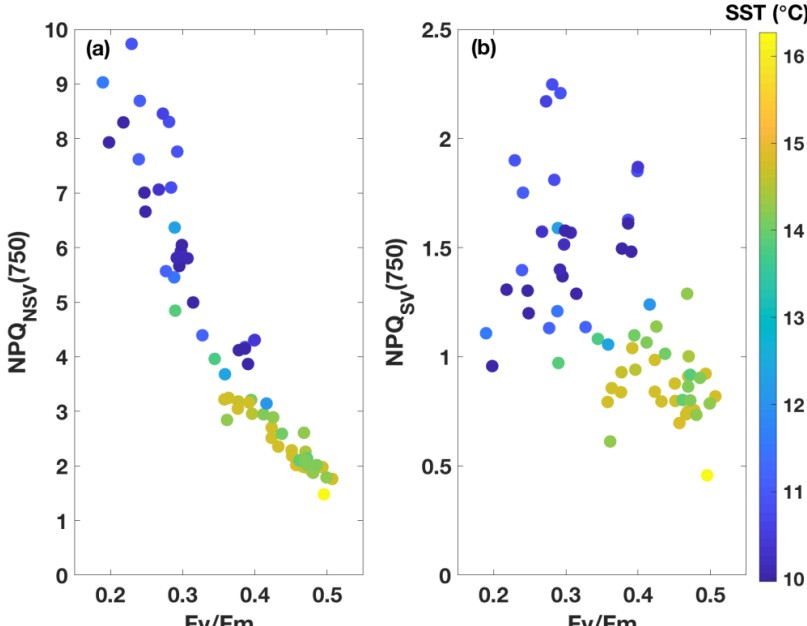

**Figure 3: FRRf results from underway samples (n=66). NPQ capacity at a light intensity of 750 μmol quanta m⁻² s⁻¹ is plotted**
**against Fv/Fm, with two different NPQ parameterizations used: Normalized Stern-Volmer (a) and Stern-Volmer (b). Colour**
**indicates SST of the corresponding waters.**





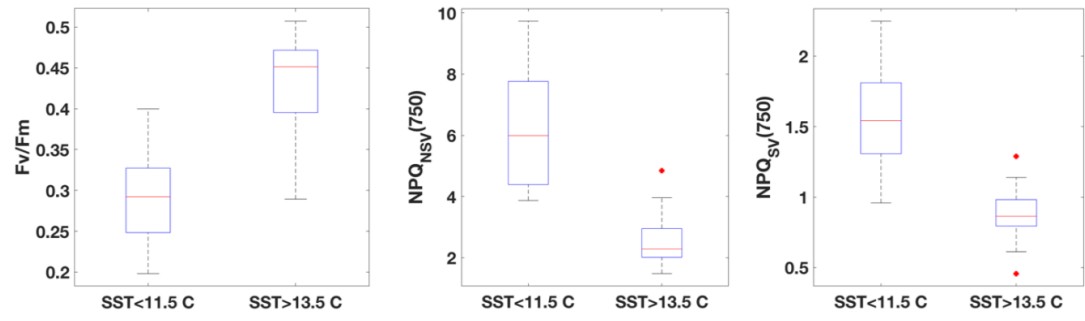


**Figure 4: Comparison of FRRf data from underway samples, grouped based on SST. All parameters are significantly different between the two respective water masses (Wilcoxon rank sum test, p<<0.01).**






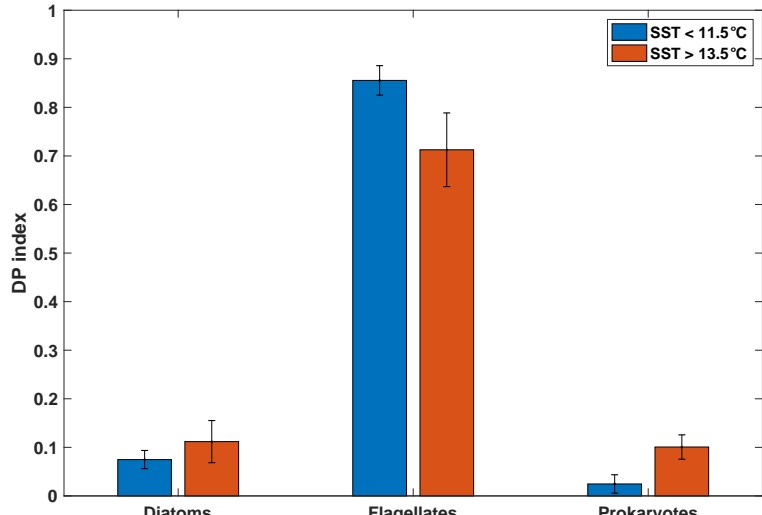

**Figure 5: Mean diagnostic pigment (DP) indices calculated based on pigment ratios from HPLC analyses for samples taken from the underway seawater supply, grouped by SST (n=6 for SST>13.5°C, n=7 for SST<11.5°C). Error bars indicate one standard deviation. See section 2.4.2 for calculation of the DP index.**







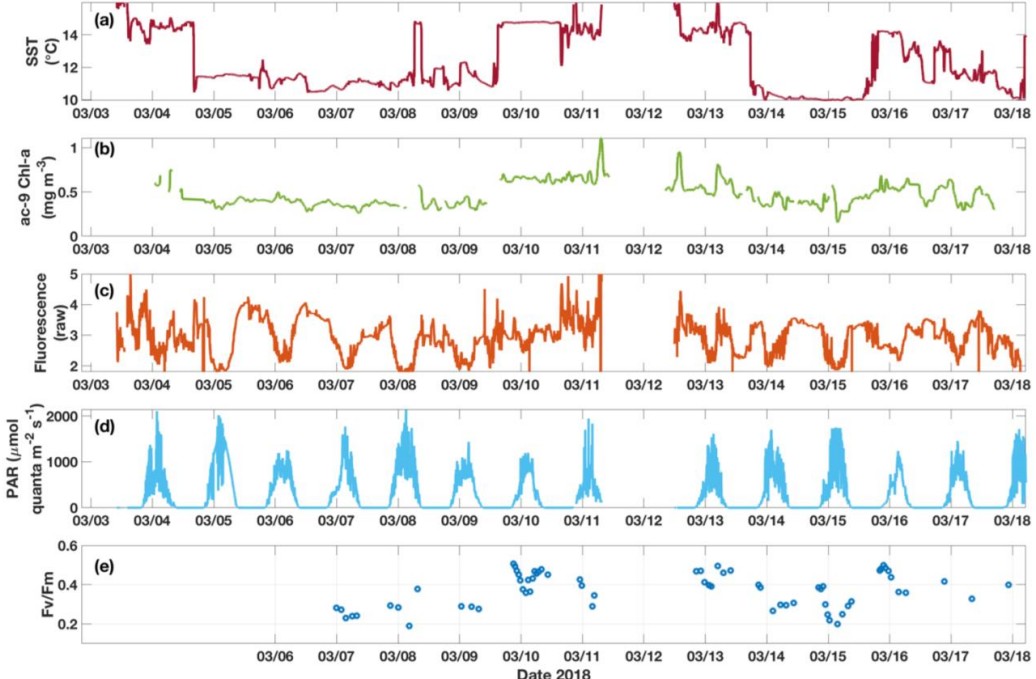

**Figure 6: Time series for underway data from the SOTS cruise in 2018 of a) SST, b) Chl-a from ac-9, c) raw fluorescence, d)**
**above-water PAR, and e) Fv/Fm measured on dark-adapted discrete samples. Diel fluctuations in fluorescence are apparent,**
**especially when cold water was sampled. These fluctuations are not found in the corresponding Chl-a time series. Note that the**
**data gap around March 12 is due to the ship leaving the SAZ region during that time.**





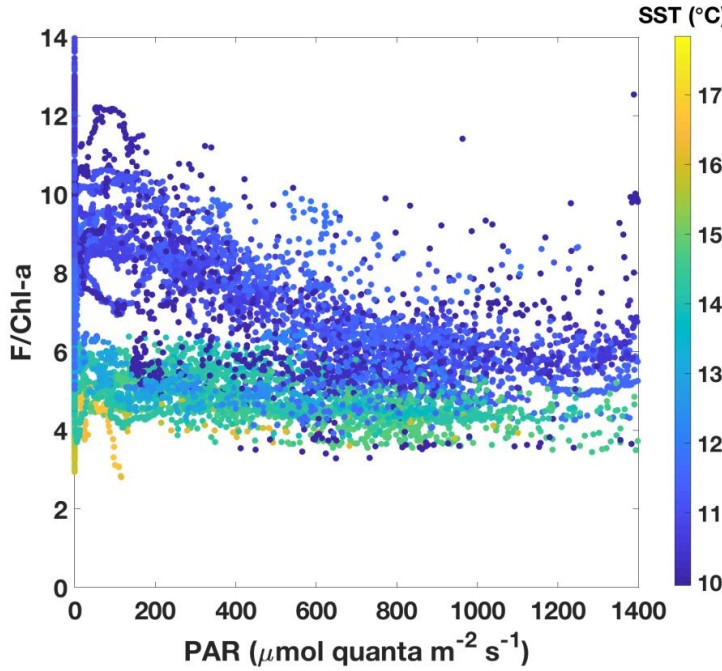


**Figure 7: Underway fluorescence data normalized to Chl-a plotted against incident PAR above the surface. Chl-a was estimated from ac-9 data. Data points are colour-coded according to SST.**






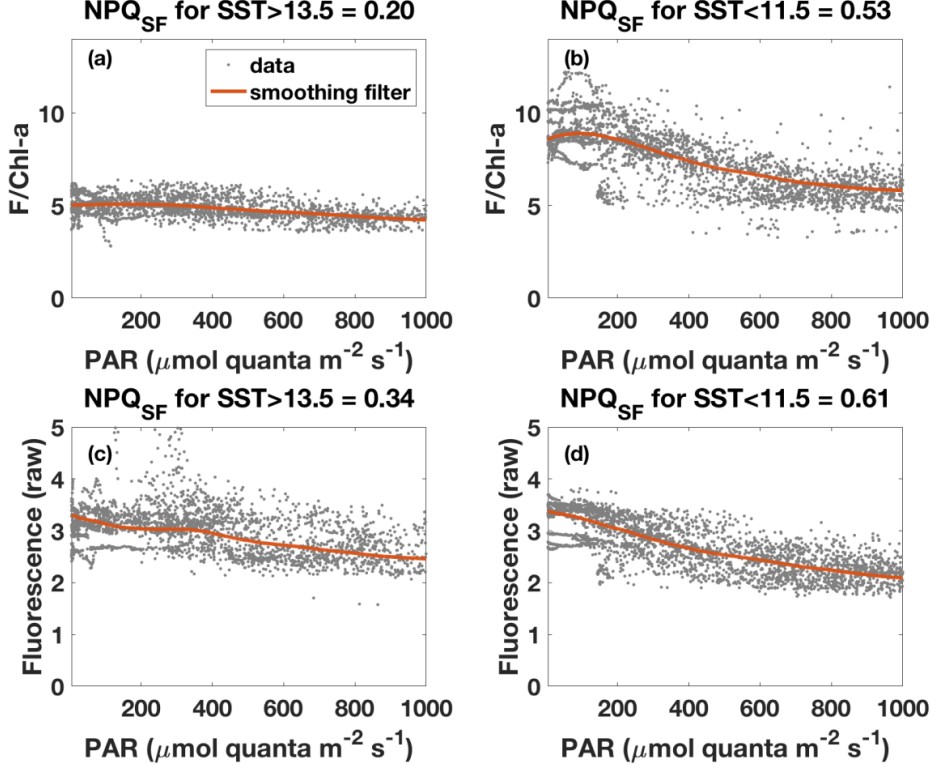

Figure 8: Fluorescence-PAR relationship as in Figure 7, but with data grouped according to the respective SST (grey dots), and with a robust Loess smoothing filter applied to the ordered data (red). Note that data for PAR $<5$ μmol quanta m$^{-2}$ s$^{-1}$ are not shown. Panels a) and b) show the results for fluorescence normalized to Chl-a (estimated based on ac-9 data), while panels c) and d) show results for unnormalized fluorescence. NPQ$_{SF}$ was calculated according to Eqn (6) using the smoothed data, with F$_{max}$ the maximum fluorescence for PAR>5 and PAR $<1000$ μmol quanta m$^{-2}$ s$^{-1}$ while F$_{min}$ was taken as the minimum fluorescence for that range.



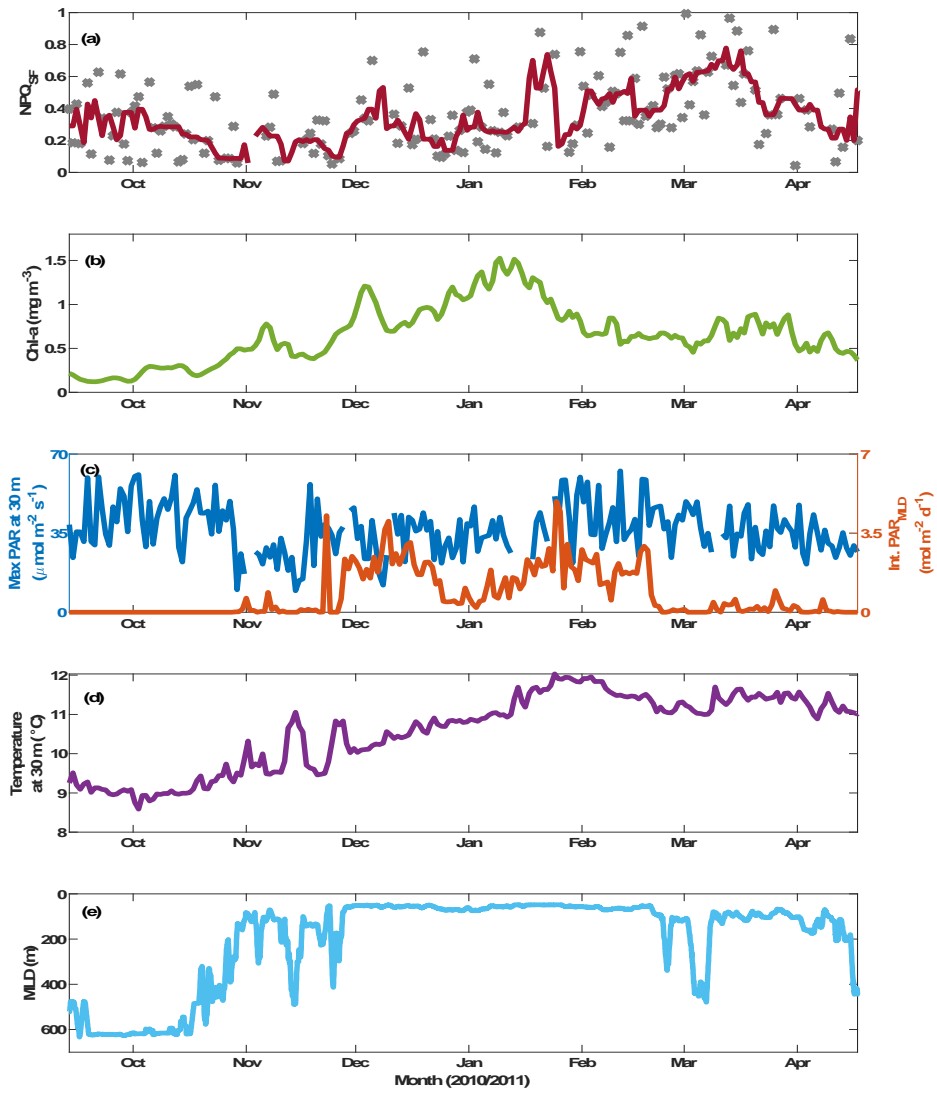


**Figure 9: SOTS mooring data for October 2010 to April 2011. Panel a) shows NPQ$_{SF}$ (see Figure S9 for a version where NPQ$_{SF}$ is normalized to PAR at F$_{min}$), with grey markers indicating daily estimates and the red line a 7-day running median. Panel b) shows a 3-point running mean over daily Chl-a concentrations estimated based on calibrated fluorescence; panel c) shows maximum**
**PAR recorded at 30 m for any given day (left axis) and the integrated daily PAR for the mixed layer (right axis). Water temperature at 30 m is shown in panel d), and the mixed layer depth is indicated in panel e) (see Section 2.6 for details).**