# Peer review of "Diel quenching of Southern Ocean phytoplankton fluorescence is related to iron limitation"

_Biogeosciences, 2019_

## Referee Comment (RC1) · Anonymous Referee #1 · 27 Sep 2019

Review of 'Diel quenching of Southern Ocean phytoplankton fluorescence is related to iron limitation' by Schallenberg et al.

This is a nice contribution that I recommend be published. I do have some comments that should be addressed and these are detailed below. Overall the manuscript is well written and the figures are clear and complete.

My main comment is related to likely potential changes in community structure in the experiments. Samples are incubated for 50-55 hours; in this time there could be significant changes in phytoplankton growth and community structure, in addition to photophysiology. For instance, a switch from being haptophyte dominated to being diatom

dominated. Therefore, without knowing the community structure/biomass data from the treatments, another explanation for the NPQ changes is that community changes are altering measured NPQ capacity between treatments, rather/in addition to the photophysiological impact of Fe availability. I think the potential for this should probably be acknowledged.

Also community structure changes are described in Fig S3 between the warm and cold waters of the transect, including the relative abundances of haptophytes and diatoms. Certainly the higher temperatures and drawdown of nitrate to 'near-zero' (line 467) would be expected to lead to community structure changes. A CHEMTAX analysis might reveal this better than the diagnostic pigment method currently shown in Fig. 5?

Section 3.2.2: Mixed layer depths between the warm and cooler regions are discussed in relation to their potential impact on photoacclimation. While differences in MLDs are discussed, really it is the light availability that is of interest for NPQ; this is a function of incident irradiance and light diffusion through the water column. Can mixed layer irradiance (mean, median), rather than mixed layer depth only, be compared between the two regions?

Paragraph starting line 455: Rates of water column integrated PP will depend not only on methodological differences (mentioned), but on the amount of phytoplankton biomass present, the temperature, and the light availability. Therefore differences in these factors, in addition to changes in Fe availability, will also affect water column integrated PP.

Line 70: Reference for the 90% value?

Section 2.2: So to be completely clear: there was only one replicate 250mL sample per treatment?

Line 174: I do not quite understand how FLCs were 'optimized to yield estimates of NPQ capacity'?

Section 2.4.3: I guess this is the non-acidification method (Holm-Hansen)?

Line 508: Can you reference figure 8 here?

---

## Referee Comment (RC2) · Anonymous Referee #2 · 28 Oct 2019

Overall this a good contribution to the literature on using chlorophyll fluorescence as a high through-put, low effort diagnostic tool and I recommend it should be published after some minor corrections. The quantification of NPQ from 'standard' fluorometers is an interesting approach and there would be fascinating work to be done applying this to the treasure trove of underway and moored fluorometer datasets both published and unpublished. I have some broader recommendations for the authors to consider, presented first, and more specific ones detailed below section-by-section.

I have some reservations about describing a 'decrease in NPQ capacity' with Fe addition (e.g. Line 336) – it gives the impression that cells are reduced capability to

[Figure]

upregulate NPQ when that is not the situation, it is that they have increased capacity for directing light to photochemistry – so perhaps a rethink of some of the language would be good. It is better discussed when described as 'increased. . . need to dissipate excess excitation energy as NPQ' (e.g. Line 344).

I agree with the point raised by Reviewer 1 regarding changes to community structure and the impact on the photophysiological parameters, especially after 55 hours of incubation. Supplementary Figure S7 suggests that Silica is limiting in the systems sampled so perhaps there may not have been a strong diatom response, and Nitrate+Nitrite is quite high so perhaps the flagellates may have remained dominant. Regardless, it needs some consideration in the results/discussion. If two populations dominated by diatoms were analysed, one iron limited and one not, would the relationship with NPQ still be as significant and reliable as a diagnostic tool? Also, what do you think was driving the increase in FvFm between the T0 and Tf in the control treatment in the experiment?

Section 1: Line 70: Reference for 90 % quenching of ChlF signal at midday?

Section 2.1 Line 137-139: Perhaps mention that FRRf measurements were done on samples from the underway also?

Section 2.2: Can you provide some justification for the iron treatments? Line 148: 'abeam' typographical error?

Line 158-159: why 51.5-55 hours? Why not subsample for FRRf measurements throughout the experiment – might have been able to see change in NPQ response before community response (if there was one)?

Section 2.3: Lines 168-174: Explanation of the light curves is confusing. Could you provide a graphical demonstration such as Figure 1 in Xu et al. 2018 doi: 10.3389/fmars.2018.00281. Also why where the duration of the light steps different? Line 168-169: 'ensure complete relaxation of all NPQ' – need to be clear here that it

is relaxation of fast-reacting/dynamic NPQ. Not all NPQ will have relaxed after 1 hour. Also, can you estimate the 'low' light intensity?

Lines 173-176: 'Optimisation for NPQ' is unclear. What are you trying to achieve here? 'Details below' is stated but I don't feel that the explanation has been provided.

Line 199: 'respective maximum light levels of the fluorescence light curves' and 'The light levels at which the NPQ parameters were measured are indicated in parentheses'. So there were different maximum light levels for different curves? Why is that and how does that impact on your comparisons of 'NPQ capacity'? Also to be clear perhaps include in the text here in parentheses the two light levels (1000 uE and 750 uE) and which experiments they correspond to.

Section 3.2.4 Line 466: 'macronutrient data from one CTD in that water mass indicate that the warm SST regime was not HNLC' – does this mean there were multiple CTDs collected in the warm water mass. It would be useful to have one more figure in the supplementary which shows position of the CTDs sampled in the 'warm' and 'cold' water masses.

Figures Figure 1: Black asterisks denoting CTD positions are hard to see. Perhaps change to another colour?

Figure 2: I think it is important to remind readers in the caption when adding in that NPQ capacity was measured at 1000 umol photons $m^2 s^{-1}$ (as stated) after 24 mins of increasing the actinic light intensity. Also, probably a good idea to include uE in the parentheses next to the light intensities to avoid confusion – active fluorometry measurements are often reported with excitation wavelength reported in parentheses next to the parameter.

---

## Author Comment (AC1) · 13 Nov 2019

We thank referee #1 for their thoughtful comments and reply to each in turn below:

Potential changes in community structure during 50-55 h incubations: In the attached Figure (which we plan to add to the Supplementary Material) we show data from Incubation 1, where subsampling took place at 29 h and 52 h (these data were not previously shown). It is evident that the changes observed at 52 h were already underway at 29 h, and in some cases had already taken place, e.g. the majority of data points for NPQSV(1000). We only subsampled this one experiment at 29 h and chose the 50-55 h incubation times for all other experiments due to the low sea water temperature (as

low as ∼10C), which would increase physiological response times compared to temperate waters. If we assume a growth rate no higher than 0.28 d-1 for the subantarctic zone (e.g., see Ryan-Keogh et al. 2018, Table 2), then the phytoplankton community would not have doubled over 50 hours. We thus conclude that 50-55 h is not enough for a significant change in the community composition, and also point to the attached figure as evidence that the overall trends found after 52 hours were generally consistent with those found at 29 h but are more pronounced. Conducting the longer incubations is thus unlikely to have biased the conclusions.

CHEMTAX vs Diagnostic Pigment Index: We chose the Diagnostic Pigment Index over CHEMTAX because it assumes less a priori knowledge specific to an oceanic region. CHEMTAX is only as good as the comparison matrix that is specified for it, and as far as we are aware, there is no available matrix specific for the Subantarctic Zone. CHEMTAX would produce a result regardless, but possibly not a very accurate one – without microscopy to match, it would be difficult to know whether the result is valid. In order to avoid overstating what our data can do, we thus decided to show the pigment composition (Figures S2 and S3), which is the raw, unbiased data, and to provide a metric for the community composition with the Diagnostic Pigment Index, while also discussing its suitability in the region and comparing our results to those found previously at the SOTS site.

Average mixed layer irradiance in the two water masses and its influence on NPQ: We disagree with the notion that the average light field is the main determinant of NPQ. When realistic vertical mixing is simulated, it appears that low-light (deep ML)-acclimated cells display a larger capacity for NPQ than high-light (shallow ML)-acclimated cells (e.g. Milligan et al. 2012). This phenomenon was likely overlooked in earlier laboratory studies because realistic mixing regimes were rarely simulated. Regardless, we are happy to include estimates of daily integrated median underwater PAR for the respective mixed layers for reference. They are calculated for the nominal date of March 30, 2018, with Kd_PAR estimated as a function of Chl following Morel

et al. (2007). The results are as follows: For the warm regime (average Chl=0.62 mg m-3 and MLD = 100 m), the median daily mixed layer PAR is 0.13 mol m-2 d-1. For average Chl = 0.38 mg m-3 and MLD=20 m and MLD=90 m as the two extremes, the median daily mixed layer PAR is 15 and 0.5 mol m-2 d-1, respectively. We propose to include these numbers in section 3.2.2 with an expanded subtitle: Mixed layer depths and average light field.

Paragraph starting line 455, Rates of water column integrated primary productivity (PP) and their dependence on phytoplankton biomass, temperature, and light availability: A comparison between the conditions reported by Westwood et al. (2007) and found in our study shows the following: Sea surface temperatures were similar (12C vs 11C, respectively), mixed layer depths were also similar (mean=38 +/- 11 m and mean=35 +/- 1 m, respectively), but the column-integrated Chl was different: mean=46 +/- 11 mg m-2 vs mean=13 mg m-2. However, the fact that biomass was lower in our study could also be the result of lower iron concentrations (and less primary productivity). Moreover, normalizing the mean column-integrated PP for each study by the mean column-integrated Chl concentration yields 85 mg C (mg Chl)-1 d-1 for Westwood et al. and 31 mg C (mg Chl)-1 d-1 for our study, indicating that the Chl-normalized PP was lower during our study (in Austral fall) than in the Westwood et al. study in Austral summer. The conclusion that PP was lower in our study thus holds, with iron limitation a probable cause, as discussed in Section 3.2.4. We plan to add the information provided above to the SI of the manuscript.

Line 70, reference for 90% depression of fluorescence by NPQ: Falkowski et al. 2017. We will include this reference in the text at line 70.

Section 2.2, only one replicate 250 mL sample per treatment for the incubations: There was indeed only one 250 mL sample per treatment in each incubation because measurement of all 4 treatments took ~2 h, so measurement of triplicates would have taken ~6 h. Given the sensitivity of fluorescence parameters to the time of day (i.e. to the light history experienced by the phytoplankton), we thought it prudent to stay in a 2-h

time window in order to be able to compare findings between treatments. We would like to point out that we grouped the three experiments, thus having produced true triplicates. We will clarify this in Section 2.2 in the updated manuscript.

Line 174, 'optimization' of FLCs for estimates of NPQ capacity: This comment is discussed in detail in the response to Referee #2.

Section 2.4.3, fluorometric Chl method: We followed the JGOFS 1994 recommendations for the 'Measurement of Chlorophyll a and Phaeopigments by Fluorometric Analysis', employing the acid ratio method described by Holm-Hansen et al. (1965). We will add this detail to the section indicated.

Line 508, add reference to Figure 8: we will add this figure reference as suggested.

References: Falkowski, P., Lin, H., and Gorbunov, M.: What limits photosynthetic energy conversion efficiency in nature? Lessons from the oceans. Philos. T. Roy. Soc. B, 372, 2-8, 2017.

Holm-Hansen, O., Lorenzen, C.J., Holmes, R.W., and Strickland, J.D.H.: Fluorometric determination of Chlorophyll, ICES J. Mar. Sci., 30, 3-15, 1965.

Milligan, A., Aparicio, U., and Behrenfeld, M.: Fluorescence and nonphotochemical quenching responses to simulated vertical mixing in the marine diatom Thalassiosira weissflogii. Mar. Ecol. Progr. S., 448, 67–78. https://doi.org/10.3354/meps09544, 2012.

Morel, A., Huot, Y., Gentili, B., Werdell, P.J., Hooker, S.B., and Franz, A.B.: Examining the consistency of products derived from various ocean color sensors in open ocean (Case 1) waters in the perspective of a multi-sensor approach, Remote Sens. Environ., 111, 69-88, 2007.

Ryan-Keogh, T.J., Thomalla, S.J., Mtshali, T.N., van Horsten, N.R., and Little, H.: Seasonal development of iron limitation in the sub-Antarctic zone. Biogeosciences, 15, 4647-4660, 2018.

Westwood, K. J., Griffiths, F. B., Webb, J. P., and Wright, S. W.: Primary production in the Sub-Antarctic and Polar Frontal Zones south of Tasmania, Australia; SAZ-Sense survey. Deep-Sea Res. Pt. II, 58, 2162–2178, 2007.

[Figure]

Supplementary Figure for response to Referee #1

[Figure]

Time course of Fv/Fm and the two NPQ parameters for incubation 1, showing the results at the end point (after 52 hours) as well as at 29 hours. Colours represent different treatments as indicated in the figure legend.

**Fig. 1.** Figure 1

---

## Author Comment (AC2) · 13 Nov 2019

We thank referee #2 for their thoughtful comments and reply to each in turn below:

Line 336: description of a 'decrease in NPQ capacity' with Fe addition: The language regarding "NPQ capacity" in this manuscript is in line with the way that Schuback and Tortell (2019) use the term, so we suggest to keep the terminology. But we do appreciate the comment about how the sentence could be misunderstood and thus propose to change the wording as follows: The NPQ capacity, measured as both NPQ_SV(1000) and NPQ_NSV(1000), decreased with iron addition, most likely due to increased capacity to use absorbed light energy for photochemistry.

Changes in community structure during 55-hour incubations: We refer to our response to referee #1. Regarding the question of whether a diatom-dominated community would have shown the same patterns: We cannot answer this question with our (haptophyte-dominated) data. However, several publications have shown similar trends to ours with respect to NPQ and Fe limitation under controlled conditions (i.e. in the laboratory or with ship-board incubations), probing a variety of species including diatoms (Schuback et al. (2015), Schuback and Tortell (2019)). The evidence thus suggests that the link between NPQ capacity and Fe status is robust across a range of phytoplankton species and thus community composition. We propose to add the information presented above to section 3.2.1.

Increase in Fv/Fm in control treatments of incubations: An increase in Fv/Fm during incubation experiments is not an uncommon feature in Fe-limited control treatments, for example see Ryan-Keogh et al. (2018). While rarely discussed in the active fluorescence literature, it is likely the result of acclimation to more constant light conditions experienced in the incubations relative to the (constantly mixing) open ocean. Less light fluctuations and thus a less variable underwater light field would allow better allocation of (limited) resources, which could result in the observed increase in Fv/Fm.

Reference for 90% quenching of ChlF signal at midday: Falkowski et al. 2017. We will include this reference in the text at line 70.

Section 2.1, Line 137-139: We will add that the underway line was also sampled for FRRf measurements as suggested.

Section 2.2: Justification for iron treatments in incubations: We chose the two different Fe treatments originally to test whether an addition of 0.2 nM would have a measurable effect (compared to the 2 nM Fe treatment which should definitely be Fe-sufficient). Since the results for both Fe treatments were consistent and we were primarily interested in the contrast between iron sufficiency vs iron limitation, we decided to bundle the +Fe treatments as described in the manuscript.

Line 148: "abeam" is not a typo but a nautical term, meaning on a line at right angles to a ship's length.

Line 158-159: Incubation duration of 51.5-55 hours and possible issues arising: See response to Referee #1 regarding community composition and the sampling that was done on Incubation 1 (including new Figure for SI).

Section 2.3, Lines 168-174: Better explanation of the light curves and what is meant by 'optimization' of FLCs for estimates of NPQ capacity (lines 173-176): The attached Figure 1 (which we suggest to add to the SI) shows the time course of light intensities in the respective FLCs. The blue line is relevant for underway samples while the red line refers to incubation samples. The different maximum light intensities were chosen such that maximum NPQ was achieved for each set of samples, while still providing an induction curve in the FRRf that cut be fitted (at higher light intensities the fluorescence induction curve becomes too flat to achieve a good fit). Incubation samples were able to handle higher light intensities (i.e. up to 1000 umol quanta m-2 s-1) than the underway samples (750 umol quanta m-2 s-1), most likely due to high-light acclimation in the incubator. Since we were interested in the NPQ capacity we chose our experimental design to maximize the NPQ in each sample set, rather than strive for uniformity. The NPQ trends observed in the incubations and underway samples are the same with respect to iron limitation, and we don't directly compare the absolute NPQ values between these two sample sets. The difference in the maximum light intensity should thus not introduce a bias in our interpretation. Regarding the length of the time steps and the 'optimization' of the FLCs: Since maximum NPQ capacity was our main focus, we chose an experimental design that struck a balance between i) ensuring an induction curve fit at the maximum light intensity, ii) increasing light levels slowly enough to allow derived fluorescence yields to reach steady state (thus choosing longer time steps at high light intensities), and iii) keeping the FLCs as short as possible so that the four treatments in each incubation experiment could be measured within the smallest time frame possible (<2h) in recognition of the sensitivity of photophysiology to light history

**BGD**

and time of day. We propose to expand lines 169-176 with a concise summary of the information provided above, while moving some of the detail about length of light steps (lines 171-172) to the Figure caption for the attached Figure 1.

Line 168-169: 'ensure complete relaxation of all NPQ': We will clarify this section to distinguish between fast-relaxing/dynamic NPQ and slow-relaxing NPQ that will not have relaxed after an hour of low-light acclimation. The light intensity during the low-light acclimation was around 2-5 umol quanta m-2 s-1, achieved by setting the sample in a small white LDPE bottle in the shaded corner of a small cooler, placed with an open lid in a dimly lit corner of the temperature-controlled laboratory.

Line 199, different maximum light levels for NPQ estimation: See response above (Section 2.3, Lines 168-174….). We will include the respective light levels and the samples they refer to on lines 199/200 as suggested.

Section 3.2.4 Line 466: 'macronutrient data from one CTD in that water mass indicate that the warm SST regime was not HNLC' – how many CTDs were done in that water mass? Only one CTD was conducted in the warm water mass and the sentence will be amended to reflect this fact as follows: '…macronutrient data from the one CTD in that water mass indicate that the warm SST regime was not HNLC'. Also see comment below re CTD locations in Figure 1.

Figure 1, CTD positions are hard to see: This Figure has been updated (attached Figure 2) to better show the CTD positions and also distinguish between CTDs in the cold water mass (overlapping on the map, cyan squares) and the one CTD in the warm water mass (red square). The figure caption will be updated also.

Figure 2: The caption to Figure 2 will be updated to include the light levels and length of FLC curves, as suggested. However, we will not include units in the parentheses of the figure labels as the current nomenclature is in line with that of Schuback and Tortell (2019), is clearly explained in Table 1, and would otherwise become even more unwieldy than it already is.

References: Falkowski, P., Lin, H., and Gorbunov, M.: What limits photosynthetic energy conversion efficiency in nature? Lessons from the oceans. Philos. T. Roy. Soc. B, 372, 2-8, 2017.

Ryan-Keogh, T.J., Thomalla, S.J., Mtshali, T.N., van Horsten, N.R., and Little, H.: Seasonal development of iron limitation in the sub-Antarctic zone. Biogeosciences, 15, 4647-4660, 2018.

Schuback, N., Schallenberg, C., Duckham, C., Maldonado, M. T., and Tortell, P. D.: Interacting effects of light and iron availability on the coupling of photosynthetic electron transport and $CO_2$-assimilation in marine phytoplankton. Plos One, 10, e0133235, https://doi.org/10.1371/journal.pone.0133235, 2015.

Schuback, N., and Tortell, P. D.: Diurnal regulation of photosynthetic light absorption, electron transport and carbon fixation in two contrasting oceanic environments. Biogeosciences, 16, 1381–1399, https://doi.org/10.5194/bg-2018-524, 2019.

Supplementary Figures for response to Referee #2

[Figure]

Figure 1: Schematic of time steps for respective light intensities in FLCs employed in this study. The blue line is relevant for underway samples (with maximum light level of 750 uM m-2 s-1) while the red line refers to incubation samples (maximum light level of 1000 uM m-2 s-1).

[Figure]

Figure 2: Updated version of Figure 1 in the manuscript, with new symbols for CTD stations: purple for CTD stations in the cold water mass and red for the one CTD station in the warm water mass.

**Fig. 1.**

---

## Author Response (AR1)

We thank referee #1 for their thoughtful comments and reply to each in turn below:

Potential changes in community structure during 50-55 h incubations:
In the attached Figure (which we plan to add to the Supplementary Material) we show data from Incubation 1, where subsampling took place at 29 h and 52 h (these data were not previously shown). It is evident that the changes observed at 52 h were already underway at 29 h, and in some cases had already taken place, e.g. the majority of data points for $NPQ_{SV}(1000)$. We only subsampled this one experiment at 29 h and chose the 50-55 h incubation times for all other experiments due to the low sea water temperature (as low as ~10C), which would increase physiological response times compared to temperate waters. If we assume a growth rate no higher than 0.28 d-1 for the subantarctic zone (e.g., see Ryan-Keogh et al. 2018, Table 2), then the phytoplankton community would not have doubled over 50 hours. We thus conclude that 50-55 h is not enough for a significant change in the community composition, and also point to the attached figure as evidence that the overall trends found after 52 hours were generally consistent with those found at 29 h but are more pronounced. Conducting the longer incubations is thus unlikely to have biased the conclusions.

CHEMTAX vs Diagnostic Pigment Index:
We chose the Diagnostic Pigment Index over CHEMTAX because it assumes less *a priori* knowledge specific to an oceanic region. CHEMTAX is only as good as the comparison matrix that is specified for it, and as far as we are aware, there is no available matrix specific for the Subantarctic Zone. CHEMTAX would produce a result regardless, but possibly not a very accurate one – without microscopy to match, it would be difficult to know whether the result is valid. In order to avoid overstating what our data can do, we thus decided to show the pigment composition (Figures S2 and S3), which is the raw, unbiased data, and to provide a metric for the community composition with the Diagnostic Pigment Index, while also discussing its suitability in the region and comparing our results to those found previously at the SOTS site.

Average mixed layer irradiance in the two water masses and its influence on NPQ:
We disagree with the notion that the average light field is the main determinant of NPQ. When realistic vertical mixing is simulated, it appears that low-light (deep ML)-acclimated cells display a larger capacity for NPQ than high-light (shallow ML)-acclimated cells (e.g. Milligan et al. 2012). This phenomenon was likely overlooked in earlier laboratory studies because realistic mixing regimes were rarely simulated.
Regardless, we are happy to include estimates of daily integrated median underwater PAR for the respective mixed layers for reference. They are calculated for the nominal date of March 30, 2018, with Kd_PAR estimated as a function of Chl following Morel et al. (2007). The results are as follows: For the warm regime (average Chl=0.62 mg m-3 and MLD = 100 m), the median daily mixed layer PAR is 0.13 mol m-2 d-1. For average Chl = 0.38 mg m-3 and MLD=20 m and MLD=90 m as the two extremes, the median daily mixed layer PAR is 15 and 0.5 mol m-2 d-1, respectively. We propose to include these numbers in section 3.2.2 with an expanded subtitle: Mixed layer depths and average light field.

Paragraph starting line 455, Rates of water column integrated primary productivity (PP) and their dependence on phytoplankton biomass, temperature, and light availability:

A comparison between the conditions reported by Westwood et al. (2007) and found in our study shows the following: Sea surface temperatures were similar (12C vs 11C, respectively), mixed layer depths were also similar (mean=38 +/- 11 m and mean=35 +/- 1 m, respectively), but the column-integrated Chl was different: mean=46 +/- 11 mg m-2 vs mean=13 mg m-2. However, the fact that biomass was lower in our study could also be the result of lower iron concentrations (and less primary productivity). Moreover, normalizing the mean column-integrated PP for each study by the mean column-integrated Chl concentration yields 85 mg C (mg Chl)-1 d-1 for Westwood et al. and 31 mg C (mg Chl)-1 d-1 for our study, indicating that the Chl-normalized PP was lower during our study (in Austral fall) than in the Westwood et al. study in Austral summer. The conclusion that PP was lower in our study thus holds, with iron limitation a probable cause, as discussed in Section 3.2.4. We plan to add the information provided above to the SI of the manuscript.

Line 70, reference for 90% depression of fluorescence by NPQ: Falkowski et al. 2017. We will include this reference in the text at line 70.

Section 2.2, only one replicate 250 mL sample per treatment for the incubations: There was indeed only one 250 mL sample per treatment in each incubation because measurement of all 4 treatments took ~2 h, so measurement of triplicates would have taken ~6 h. Given the sensitivity of fluorescence parameters to the time of day (i.e. to the light history experienced by the phytoplankton), we thought it prudent to stay in a 2-h time window in order to be able to compare findings between treatments. We would like to point out that we grouped the three experiments, thus having produced true triplicates. We will clarify this in Section 2.2 in the updated manuscript.

Line 174, 'optimization' of FLCs for estimates of NPQ capacity:
This comment is discussed in detail in the response to Referee #2.

Section 2.4.3, fluorometric Chl method: We followed the JGOFS 1994 recommendations for the 'Measurement of Chlorophyll a and Phaeopigments by Fluorometric Analysis', employing the acid ratio method described by Holm-Hansen et al. (1965). We will add this detail to the section indicated.

Line 508, add reference to Figure 8: we will add this figure reference as suggested.

We thank referee #2 for their thoughtful comments and reply to each in turn below:

Line 336: description of a 'decrease in NPQ capacity' with Fe addition:
The language regarding "NPQ capacity" in this manuscript is in line with the way that Schuback and Tortell (2019) use the term, so we suggest to keep the terminology. But we do appreciate the comment about how the sentence could be misunderstood and thus propose to change the wording as follows:
The NPQ capacity, measured as both NPQ_SV(1000) and NPQ_NSV(1000), decreased with iron addition, most likely due to increased capacity to use absorbed light energy for photochemistry.

Changes in community structure during 55-hour incubations:
We refer to our response to referee #1. Regarding the question of whether a diatom-dominated community would have shown the same patterns: We cannot answer this question with our (haptophyte-dominated) data. However, several publications have shown similar trends to ours with respect to NPQ and Fe limitation under controlled conditions (i.e. in the laboratory or with ship-board incubations), probing a variety of species including diatoms (Schuback et al. (2015), Schuback and Tortell (2019)). The evidence thus suggests that the link between NPQ capacity and Fe status is robust across a range of phytoplankton species and thus community composition. We propose to add the information presented above to section 3.3.1.

Increase in Fv/Fm in control treatments of incubations:
An increase in Fv/Fm during incubation experiments is not an uncommon feature in Fe-limited control treatments, for example see Ryan-Keogh et al. (2018). While rarely discussed in the active fluorescence literature, it is likely the result of acclimation to more constant light conditions experienced in the incubations relative to the (constantly mixing) open ocean. Less light fluctuations and thus a less variable underwater light field would allow better allocation of (limited) resources, which could result in the observed increase in Fv/Fm.

Reference for 90% quenching of ChlF signal at midday:
Falkowski et al. 2017. We will include this reference in the text at line 70.

Section 2.1, Line 137-139: We will add that the underway line was also sampled for FRRf measurements as suggested.

Section 2.2: Justification for iron treatments in incubations:
We chose the two different Fe treatments originally to test whether an addition of 0.2 nM would have a measurable effect (compared to the 2 nM Fe treatment which should definitely be Fe-sufficient). Since the results for both Fe treatments were consistent and we were primarily interested in the contrast between iron sufficiency vs iron limitation, we decided to bundle the +Fe treatments as described in the manuscript.

Line 148: "abeam" is not a typo but a nautical term, meaning on a line at right angles to a ship's length.

Line 158-159: Incubation duration of 51.5-55 hours and possible issues arising:
See response to Referee #1 regarding community composition and the sampling that was done on Incubation 1 (including new Figure for SI).

Section 2.3, Lines 168-174: Better explanation of the light curves and what is meant by 'optimization' of FLCs for estimates of NPQ capacity (lines 173-176):
The attached Figure 1 (which we suggest to add to the SI) shows the time course of light intensities in the respective FLCs. The blue line is relevant for underway samples while the red line refers to incubation samples. The different maximum light intensities were chosen such that maximum NPQ was achieved for each set of samples, while still providing an induction curve in the FRRf that cut be fitted (at higher light intensities the fluorescence

induction curve becomes too flat to achieve a good fit). Incubation samples were able to handle higher light intensities (i.e. up to 1000 umol quanta m-2 s-1) than the underway samples (750 umol quanta m-2 s-1), most likely due to high-light acclimation in the incubator. Since we were interested in the NPQ capacity we chose our experimental design to maximize the NPQ in each sample set, rather than strive for uniformity. The NPQ trends observed in the incubations and underway samples are the same with respect to iron limitation, and we don't directly compare the absolute NPQ values between these two sample sets. The difference in the maximum light intensity should thus not introduce a bias in our interpretation.

Regarding the length of the time steps and the 'optimization' of the FLCs: Since maximum NPQ capacity was our main focus, we chose an experimental design that struck a balance between i) ensuring an induction curve fit at the maximum light intensity, ii) increasing light levels slowly enough to allow derived fluorescence yields to reach steady state (thus choosing longer time steps at high light intensities), and iii) keeping the FLCs as short as possible so that the four treatments in each incubation experiment could be measured within the smallest time frame possible (<2h) in recognition of the sensitivity of photophysiology to light history and time of day.

We propose to expand lines 169-176 with a concise summary of the information provided above, while moving some of the detail about length of light steps (lines 171-172) to the Figure caption for the attached Figure 1.

Line 168-169: 'ensure complete relaxation of all NPQ':
We will clarify this section to distinguish between fast-relaxing/dynamic NPQ and slow-relaxing NPQ that will not have relaxed after an hour of low-light acclimation. The light intensity during the low-light acclimation was around 2-5 umol quanta m-2 s-1, achieved by setting the sample in a small white LDPE bottle in the shaded corner of a small cooler, placed with an open lid in a dimly lit corner of the temperature-controlled laboratory.

Line 199, different maximum light levels for NPQ estimation:
See response above (Section 2.3, Lines 168-174….). We will include the respective light levels and the samples they refer to on lines 199/200 as suggested.

Section 3.2.4 Line 466: 'macronutrient data from one CTD in that water mass indicate that the warm SST regime was not HNLC' – how many CTDs were done in that water mass?
Only one CTD was conducted in the warm water mass and the sentence will be amended to reflect this fact as follows: '…macronutrient data from the one CTD in that water mass indicate that the warm SST regime was not HNLC'. Also see comment below re CTD locations in Figure 1.

Figure 1, CTD positions are hard to see:
This Figure has been updated (attached Figure 2) to better show the CTD positions and also distinguish between CTDs in the cold water mass (overlapping on the map, cyan squares) and the one CTD in the warm water mass (red square). The figure caption will be updated also.

Figure 2: The caption to Figure 2 will be updated to include the light levels and length of FLC curves, as suggested. However, we will not include units in the parentheses of the figure

labels as the current nomenclature is in line with that of Schuback and Tortell (2019), is clearly explained in Table 1, and would otherwise become even more unwieldy than it already is.

[revised manuscript text omitted]

-

[Figure]

Figure S1: Time course of Fv/Fm as well as the two NPQ parameters for incubation 1, showing the results at the end point (after 52 h) as well as at 29 h. Colors represent different treatments as indicated in the figure legend. It is evident that the changes observed at 52 h were already underway at 29 h, but were generally more pronounced after 52 h.

[Figure]

Figure S2: Schematic of time steps and light levels in fluorescence light curves employed in this study. The blue line relates to underway samples (with maximum light level of 750 μmol quanta m$^{-2}$ s$^{-1}$) and the red line refers to samples from the deckboard incubation with maximum light levels of 1000 μmol quanta m$^{-2}$ s$^{-1}$. The different maximum light intensities were chosen such that maximum NPQ was achieved for each set of samples, while still providing an induction curve in the FRRf that could be readily fit to the expected functionality (at higher light intensities the curve becomes too flat to achieve a good fit). Phytoplankton in the incubations displayed variable fluorescence at higher light intensities than phytoplankton in the underway samples, most likely due to high-light acclimation in the incubated samples.

[Figure]

Figure S3S1: Comparison of Chl-a estimates from fluorometric analyses and HPLC. Linear regression yields the following expression: $Chl_{HPLC} = Chl_F * 0.879 - 0.052$ ($r^2 = 0.97$, n = 15, standard error of the estimate = 0.029 mg m$^{-3}$).

[Figure]

Figure S42: Initial pigment concentrations relative to Chl-a from HPLC analyses for the three incubation experiments. The sea surface temperature at time of sampling is indicated at the top. The HPLC analyses indicate that the sampled waters were dominated by haptophytes. S, and some diatoms and chrysophytes may also have been present.

[Figure]

Figure S53: Pigment concentrations relative to Chl-a from HPLC analyses  of underway samples, grouped by SST. The HPLC analyses indicate that the cold waters were dominated by haptophytes, and some diatoms and chrysophytes may also have been present. Haptophytes, as indicated by the presence of Hex-fuco, were also present in the warm water mass, but they were less dominant here than in the cold waters. The warm waters also hosted some green algae (prasinophytes) and likely some diatoms, and some cyanobacteria may have been present, most likely *Synechococcus*. The data suggest that both the warm and cold waters had mostly similar phytoplankton sizes, except for the possibility of *Synechococcus* in the warm waters.

[Figure]

Figure S64: Mean phytoplankton absorption spectra for the respective water masses (indicated byred = warm; blue = and cold colours) from underway samples (n=17), with one standard deviation indicated by shaded areas. Top panel shows phytoplankton absorption spectra normalized to the respective Chl-a concentrations, while the bottom panel shows phytoplankton absorption spectra normalized to the mean absorption $\langle a_{phyto} \rangle$.

[Figure]

Figure S75: Temperature and density profiles from CTD casts on the SOTS voyage in 2018. The red line is the only profile from the warm water mass (SST>13.5°C), and the dark green profile was likely measured in a transitional zone. All other profiles are from the cold water mass and show varying mixed layer depths.

[Figure]

Figure S8̶6̶: Individual fluorescence parameters measured with the FRRf on underway samples, normalized to Chl-a and grouped by SST. The minimum and maximum fluorescence in the dark-adapted state ($F_o$ and $F_m$) are shown in the top panels, while the respective measurements in the light-acclimated state (in this case at 750 μmol quanta $m^{-2}$ $s^{-1}$) are shown at the bottom. All parameters are significantly different ($p<<0.01$) between the two water masses.

[Figure]

Figure S97: Profiles of macronutrient concentrations measured on the SOTS voyage in 2018, colour-coded by the respective temperature. Only one profile was measured in the warm water mass (yellow colours at the surface), exhibiting significantly lower $NO_x$ and $PO_4$ concentrations than the colder water mass.

[Figure]

Figure S108: Temperature-salinity diagram for CTD data from the SOTS 2018 voyage. The purple line indicates data from the water mass with a warmer SST signature. This water mass also exhibited a saltier salinity minimum at depth, consistent with this water mass originating further north than the  Subantarctic cold water mass (Herraiz-Borreguero & Rintoul, 2011).

[Figure]

Figure S119: SOTS mooring data for October 2010 to April 2011; same as Figure 9 in the main manuscript except for panel A. Here, panel A shows $NPQ_{SF}$ normalized to PAR at $F_{min}$, with grey markers indicating daily estimates and the red line a 7-day running median. Panel B shows a 3-point running mean over daily Chl-a concentrations estimated based on calibrated fluorescence; panel C shows maximum PAR recorded at 30 m for any given day (left axis) and the integrated daily PAR for the mixed layer (right axis). Water temperature at 30 m is shown in panel D, and the mixed layer depth is indicated in panel E.

**Comparison of conditions underlying measurements of primary productivity.**

Our discussion in Section 3.2.4 compares findings regarding primary productivity between our study and that of Westwood et al. (2011). A number of factors can contribute to the observed differences in primary productivity, hence we compare the conditions found in the two studies in order to pinpoint the most likely candidate. Sea surface temperatures were similar between Westwood et al. (2011) and our study (12°C vs 11°C, respectively), mixed layer depths were also similar (mean=38 +/- 11 m and mean=35 +/- 1 m, respectively), but the column-integrated Chl-a was different: mean=46 +/- 11 mg m$^{-2}$ vs mean=13 mg m$^{-2}$. However, the fact that biomass was lower in our study could also be the result of lower iron concentrations (and lower rates of primary productivity). Moreover, normalizing the mean column-integrated primary productivity for each study by the mean column-integrated Chl-a concentration yields 85 mg C (mg Chl)$^{-1}$ d$^{-1}$ for Westwood et al. (2011) and 31 mg C (mg Chl)$^{-1}$ d$^{-1}$ for our study, indicating that the Chl-normalized primary productivity was lower during our study (in Austral fall) than in the Westwood et al. study in Austral summer. The conclusion that primary productivity was lower in our study thus holds, with Fe limitation a probable cause, as discussed in Section 3.2.4.

**Treatment and calibration of ac-9 data:**

The largest problem encountered on the cruise was the presence of bubbles in the flow tubes, as has been reported previously (Slade et al., 2010). This problem was intermittent, i.e. there were whole multi-hour stretches without bubbles, but overall the problem persisted through the duration of the cruise. In order to make the most of the data, the raw data (file by file and wavelength by wavelength, both filtered and unfiltered) were processed as follows:

[revised manuscript text omitted]